# COMPOSITION-GROUNDED DATA SYNTHESIS FOR VISUAL REASONING

**Xinyi Gu**[1][*] **Jiayuan Mao**[1] **Zhang-Wei Hong**[1] **Zhuoran Yu**[2] **Pengyuan Li**[3] **Dhiraj Joshi**[3] **Rogerio Feris**[3] **Zexue He**[1,3]
[1]MIT   [2]UW-Madison   [3]MIT-IBM Watson AI Lab

## ABSTRACT

Pretrained multi-modal large language models (MLLMs) demonstrate strong performance on diverse multimodal tasks, but remain limited in reasoning capabilities for domains where annotations are difficult to collect. In this work, we focus on artificial image domains such as charts, rendered documents, and webpages, which are abundant in practice yet lack large-scale human annotated reasoning datasets. We introduce COGS (COmposition-Grounded data Synthesis), a data-efficient framework for equipping MLLMs with advanced reasoning abilities from a small set of seed questions. The key idea is to decompose each seed question into primitive perception and reasoning *factors*, which can then be systematically recomposed with new images to generate large collections of synthetic question-answer pairs. Each generated question is paired with subquestions and intermediate answers, enabling reinforcement learning with factor-level process rewards. Experiments on chart reasoning show that COGS substantially improves performance on unseen questions, with the largest gains on reasoning-heavy and compositional questions. Moreover, training with a factor-level mixture of different seed data yields better transfer across multiple datasets, suggesting that COGS induces generalizable capabilities rather than dataset-specific overfitting. We further demonstrate that the framework extends beyond charts to other domains such as webpages. Project page: https://cogsynthesis.github.io.

## 1 INTRODUCTION

Pretrained multi-modal large language models (MLLMs) have achieved impressive performance across a wide range of multimodal tasks (Liu et al., 2023c; Bai et al., 2025; Wang et al., 2025a; Agrawal et al., 2024; OpenAI et al., 2024; Comanici et al., 2025; Anthropic, 2024), yet advanced reasoning capabilities remain underdeveloped, especially in domains where user reasoning-intensive query-answer data is difficult to collect. In this work, we consider reasoning capability over artificial image domains, including charts, tables, information graphs, rendered documents, webpages, etc. While such images are abundant on the web, datasets containing reasoning questions over them are scarce. However, developing MLLMs that can handle these reasoning queries is critical for downstream applications, such as building agents that interpret and edit documents or take actions in digital environments.

In this paper, we aim to equip MLLMs with these missing capabilities using only a small set of *seed questions* in a target domain. Our goal is to bootstrap from these seed questions to generate a large, diverse dataset of synthetic question-answer pairs, leveraging additional unlabeled images such as online charts or webpages. The key insight of our approach is *compositionality*: although a seed question set may contain only a limited number of surface forms, each question can be decomposed into a set of smaller subquestions, which we call *factors* in this paper. Factors may capture primitive perception and reasoning steps, such as reading a number from a chart, comparing two entries in a table, or performing arithmetic. Crucially, these factors can be recombined systematically and compositionally to produce a much larger space of complex questions.

We propose a data-efficient framework, COGS (COmposition-Grounded data Synthesis), that operationalizes this idea in three stages. First, given a seed dataset of questions in the target domain, we

---

[*]Work done during an internship at MIT-IBM Watson AI Lab. Corresponding to `gxy@mit.edu`, `zexueh@mit.edu`.

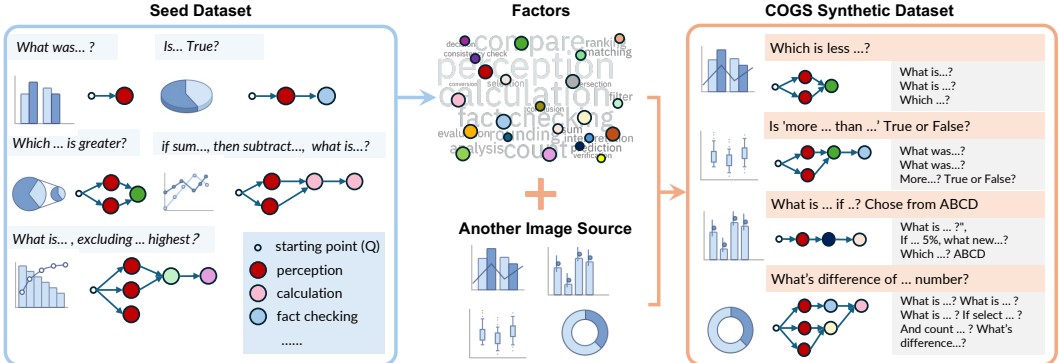

Figure 1: COGS: Starting from a small set of reasoning-intensive seed questions, COGS decomposes them into primitive perception and reasoning factors, which are then recombined with new image sources to synthesize question–answer pairs. This process expands both the quantity and diversity of reasoning types beyond the original seeds. Fig. 2 shows an illustrative example.

decompose each question into its constituent perception and reasoning factors. Second, we recombine subsets of discovered factors with new images to generate novel compositional questions, each paired with intermediate subquestions and intermediate answers as a bonus. Finally, we finetune a pretrained MLLM using Group Relative Policy Optimization (GRPO; Shao et al., 2024), augmented with process rewards derived from the factor annotations for fine-grained supervision.

We begin our evaluation on chart reasoning, a domain that exemplifies the scarcity of annotated reasoning questions despite the ubiquity of images. Our experiments show that COGS significantly improves the reasoning capabilities of base MLLMs, with the largest gains observed on reasoning-heavy and compositional questions. Moreover, our framework naturally supports mixtures of datasets: training jointly on multiple datasets yields positive transfer, demonstrating that the model acquires transferable capabilities rather than overfitting to a particular dataset. Finally, we illustrate that the same framework extends to the webpage reasoning domain, highlighting its broad applicability.

In summary, this work introduces a principled approach to bootstrapping new reasoning skills in pretrained MLLMs from a small seed query set, by exploiting the factorized structure of questions to unlock scalable synthetic data generation and process-level reinforcement learning.

## 2 RELATED WORK

Understanding artificial image such as charts and web GUIs demands grounded perception and substantial visual reasoning. General-purpose MLLMs make this feasible through large-scale pre-training and instruction tuning (Liu et al., 2023c; Bai et al., 2025; Wang et al., 2025a; Agrawal et al., 2024; OpenAI et al., 2024; Comanici et al., 2025; Anthropic, 2024). Prior work on automatic instruction generation (Wang et al., 2023c; Yuan et al., 2024), refinement, and evolutionary methods (Xu et al., 2024a; Zeng et al., 2024) increases the complexity of synthetic data for text reasoning. While these methods mainly search for reasoning trajectories in text space, we aim to analyze and augment the seed dataset by automatically detecting reasoning components grounded in visual features. Unlike approaches that rely on hand-crafted heuristics (Xu et al., 2024a) or strong pretrained language models (Zeng et al., 2024), COGS extracts component groups from the seed data and uses them to customize the dataset for the target task. In parallel with generalist MLLMs, specialist models have been introduced to target these domains more directly, prioritizing structured text extraction, numeric grounding, and compositional reasoning. New benchmarks and data-synthesis methods based on human-defined heuristics have followed, and specialist models have been trained accordingly.

**Chart understanding** including description and reasoning tasks. Benchmarks emphasize diverse image sources have been designed with different templates or task formulation to evaluate the models performance (Kahou et al., 2018; Kafle et al., 2018; Methani et al., 2020; Masry et al., 2022; Xu et al., 2024b; Xia et al., 2025; Shi et al., 2024; He et al., 2024). Recent human-curated evaluations emphasize scientific or real figures, multi-chart settings, and open-vocabulary answers, and

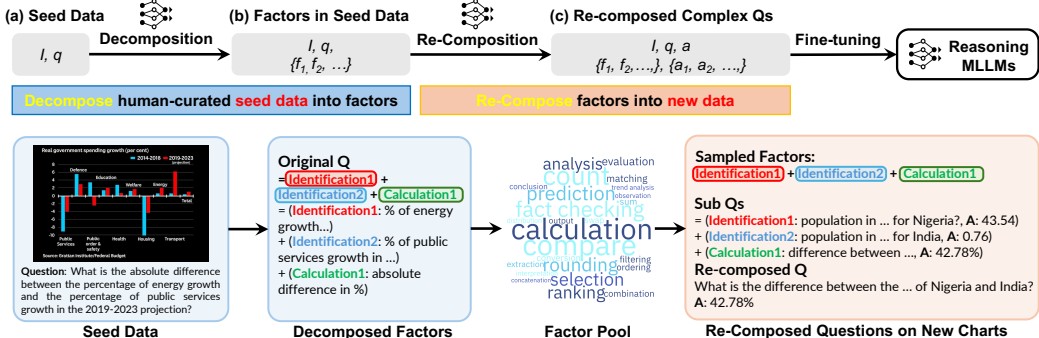

Figure 2: The framework of COGS consists of three stages: seed question decomposition, factor recomposition, and model fine-tuning.

they raise the bar with inference-heavy questions that demand reasoning before answering (Masry et al., 2025a; Liu et al., 2024a; Wang et al., 2024b; Tang et al., 2025; Huang et al., 2025). Datasets and training strategies to specialist a MLLM have been developed. Specialist pipeline methods first convert charts into structured intermediates and utilize an LLM answer on top of that representation, which improves numerical fidelity when extraction is accurate (Lee et al., 2023; Liu et al., 2023b;a; Xia et al., 2025). Specialist model can also be trained end-to-end for unify perception and reasoning inside a VLM, often by aligning multi-format inputs and instruction-tuning (Han et al., 2023; Carbune et al., 2024; Meng et al., 2024; Ye et al., 2023b;c; 2024; Chen et al., 2023; Hu et al., 2024; Liu et al., 2024c; Wang et al., 2023a; Liu et al., 2024a; Zhang et al., 2024c; Yan et al., 2024; Masry et al., 2025b; Chen et al., 2024; Zhao et al., 2025; Jia et al., 2025; Wu et al., 2025; Xu et al., 2025). Data synthesis approaches including question-level template and in-context example has been designed for specialist fine-tune a pretrained model (Li et al., 2024b; Chen et al., 2025; He et al., 2023).

**GUI understanding.** GUI understanding has been studied through a growing set of benchmarks for page comprehension and reasoning over website and app UIs (Awal et al., 2025; Liu et al., 2024b; Chen et al., 2021; Hsiao et al., 2025; Li et al., 2020; Chang et al., 2022; Wang et al., 2024a), as well as for grounding and agentic predictions (Liu et al., 2024b; Li et al., 2025; Cheng et al., 2024). Models have been developed for specialist tasks such as information extraction (Baek et al., 2019), detection/localization of UI elements for agentic use (Hu et al., 2024; Lee et al., 2023; Hong et al., 2024; Zheng et al., 2024; Gou et al., 2025) and general-purpose UI reasoning. (Ye et al., 2023a; Baechler et al., 2024; You et al., 2024; Liu et al., 2025; Wang et al., 2025b).

## 3 COGS

Fig. 2 provides an overview of COGS, which consists of three stages. First, given a seed dataset of questions in the target domain, we decompose each question into its underlying perception and reasoning factors. Second, the framework collects all discovered factors across the domain and, together with an image collection, generates new questions by recomposing a randomly sampled subset of these factors. Finally, the newly generated questions are used to finetune a pretrained MLLM. During this stage, we leverage the factor decompositions associated with each generated question to define process-level rewards.

In this section, we begin with the problem formulation in Section 3.1, then describe factor decomposition in Section 3.2 and question generation via factor recomposition in Section 3.3, and finally discuss the process reward design for RL finetuning in Section 3.4. Prompts for decomposition and recomposition are presented in Appendix E.

### 3.1 PROBLEM FORMULATION

Our goal is to fine-tune a multimodal large language model (MLLM) to acquire new capabilities in answering complex, compositional questions in a target domain. Let $\mathcal{Q}$ denote the set of natural language questions in this domain, and let $\mathcal{I}$ denote the corresponding set of images. A question $q \in \mathcal{Q}$ can often be interpreted as requiring a sequence of *perception factors* (e.g., identifying a number in a chart or localizing an element by its relation to another element on a webpage) and *reasoning factors* (e.g., logic, arithmetic, or spatial reasoning). We denote the factorized representation of a question as $q \mapsto \{f_1, f_2, \ldots, f_k\}, \quad f_i \in \mathcal{F}$, where $\mathcal{F}$ is the set of possible factors.

Our objective is to use a small *seed question dataset* $\mathcal{Q}^0$ to bootstrap a process that can (i) discover the relevant set of factors $\mathcal{F}$ in the target domain, (ii) generate novel and valid questions by recomposing subsets of factors, and (iii) use these generated questions to improve a pretrained MLLM through reinforcement learning. Notably, we do not require ground-truth answers for the seed questions, which makes the data collection process more scalable.

## 3.2 Seed Data Decomposition

The first stage of our framework COGS is the decomposition of seed questions into a set of interpretable factors. As illustrated in Fig. 2, a complex question that asks for the energy growth and public service growth can be broken down into distinct *perception factors* and *reasoning factors*. In this example, the question requires (i) identifying the percentage of energy growth (`Perception1`), (ii) identifying the percentage of public services growth (`Perception2`), and (iii) computing their absolute difference (`Calculation1`).

We obtain such decompositions by prompting a MLLM. Specifically, we provide the MLLM with a natural language description of the decomposition task, a set of in-context examples (each consists of a paired question and its list of factors), the target question to be decomposed, and the image associated with this question to ensure each factor is visual-grounded. This step essentially recovers the factorized representations of the given question $q \mapsto \{f_1, f_2, \ldots, f_k\}, \quad f_i \in \mathcal{F}$. For each factor, the MLLM outputs a category label (e.g., `Calculation`, `Counting`) and a corresponding *subquestion* that describes the role of this factor in the original question. These subquestions serve as exemplars of target categories that will later be used during the factor recomposition stage.

We then aggregate all factors discovered from $\mathcal{Q}^0$ to form the space of possible factors $\mathcal{F}$. Each factor is represented by a category name (e.g., `Calculation`, `Counting`, `Comparison`) and is associated with a set of exemplar subquestions extracted from the seed dataset. The obtained factor set $\mathcal{F}$ serves two purposes. First, it builds a compositional representation of the latent structure underlying complex questions, making it possible to recombine factors into new questions in the domain. Second, it provides fine-grained supervision for reinforcement learning: since each generated question is associated with its underlying factors, we can define process rewards that provide intermediate signals for accomplishing individual reasoning steps.

## 3.3 Question Generation via Factor Recomposition

The second stage of our framework COGS is to generate new questions by recomposing previously discovered factors. As illustrated in Fig. 2, the input to this stage includes: (i) a textual description of the recomposition task together with a single question recomposition example, (ii) a new image $I$ from any source, (iii) a list of factors subsampled from $\mathcal{F}$. Each factor is specified by its category name and a sampled subset of subquestions from the seed dataset $\mathcal{Q}^0$.

We prompt a MLLM with this input to generate new subquestions of similar kinds but grounded on the new image. The MLLM then composes these subquestions into a coherent overall question. Alongside question generation, the MLLM is also responsible for producing answers: answers to subquestions are generated first, which are then combined to form the answer to the recomposed overall question. Therefore, the generated data pairs consist of both the overall question-answer pair $(q, a)$ and its associated factor-level subquestions and answers. Formally, each data point is as a tuple $\langle I, q, a, \{f_i\}, \{a_i\}\rangle$ where $q \mapsto \{f_1, f_2, \ldots, f_k\}$ and $a_i = Answer(f_i \mid I)$.

An additional advantage arises in artificial domains such as charts, where images are often accompanied by underlying metadata (e.g., tables of data associated with the figures). In such cases, we leverage this auxiliary metadata during question generation to improve answer precision. This idea is consistent with prior work in synthetic data generation for structured domains (Masry et al., 2022).

Overall, this recomposition procedure enables us to expand the training distribution compositionally, generating diverse questions grounded solely from a dataset of unlabeled images without requiring additional question–answer annotations.

## 3.4 Reinforcement Learning-Based Fine-tuning

The final stage of COGS is reinforcement learning fine-tuning, where we adopt Group Relative Policy Optimization (GRPO; Shao et al., 2024) to fine-tune a pretrained MLLM with the generated question–answer data. A key advantage of our recompositional design is that each complex ques-

tion is automatically paired with its corresponding subquestions and sub-answers during the data generation phase. This structure enables richer reward modeling beyond final-answer correctness.

In RL-based fine-tuning for MLLMs, the most common choice of reward model is to assign rewards based on exact or approximate answer matching (e.g., F1 string score). However, since COGS generates both overall questions and their factor-level subquestions, we can define additional *process rewards* that assess whether intermediate reasoning steps are carried out correctly. Concretely, given a data point $\langle I, q, a, \{f_i\}, \{a_i\} \rangle$, for each factor $f_i$ with subquestion $s_i$ and ground-truth answer $a_i$, we prompt an LLM-based reward model to verify whether the model's chain-of-thought reasoning produced the correct sub-answer. This yields a binary score $c_i \in \{0, 1\}$ for each factor.

Formally, let $r^{\text{final}}(y) \in \{0, 1\}$ denote the correctness of the final answer for output $y$, $N$ the number of subquestions, and $\lambda > 0$ a weighting hyperparameter. We define the subquestion hit rate as $r^{\text{sub}}(y) = \frac{1}{N} \sum_{i=1}^{N} c_i$. In this work, we consider three reward models:

- **StandardRM**: $r(y) = r^{\text{final}}(y)$, which only evaluates final-answer correctness. This is the default option when subquestion supervision is not available.
- **ProcessRM-sum**: $r(y) = r^{\text{final}}(y) + \lambda \cdot r^{\text{sub}}(y)$, which combines correctness of the final answer with the average subquestion accuracy, encouraging faithful reasoning at the factor level.
- **ProcessRM-max**: $r(y) = \max\left(r^{\text{final}}(y), \lambda \cdot r^{\text{sub}}(y)\right)$, which prioritizes the final answer but still provides reward shaping when the intermediate reasoning is largely correct.

The summation-based process reward is a common choice. However, because a question may admit multiple valid decompositions and the resulting factor-level signals are noisy, the summed reward can misrank policies. By contrast, ProcessRM-max preserves policy orders. In contrast, our analysis shows that the max-based reward is order-preserving with respect to final-answer accuracy.

**Proposition 3.1** *Assume $r^{\text{final}} \in \{0, 1\}^*$, $\lambda \in (0, 1)$ and $r^{\text{sub}} \in [0, 1]$. $r^{\text{sub}}$ is a noisy shaping reward: $r^{\text{sub}} = \alpha\, r^{\text{final}} + \varepsilon$ and with $\alpha \in [0, 1]$. Note that $\mathbb{E}_\pi[\varepsilon]$ may vary with $\pi$. Define $V_f(\pi) = \mathbb{E}_\pi[r^{\text{final}}]$*

- *ProcessRM-max preserves policy orders. For any policies $\pi_1, \pi_2$,*

$$\text{sign}\left(V_f(\pi_1) - V_f(\pi_2)\right) = \text{sign}\left(\mathbb{E}[r^{\max} | \pi_1] - \mathbb{E}[r^{\max} | \pi_2]\right),$$

- *ProcessRM-sum does not necessarily preserve policy orders. That is, there exist policies $\pi_1, \pi_2$ with $V_f(\pi_1) > V_f(\pi_2)$, $\mathbb{E}[r^\Sigma | \pi_1] - \mathbb{E}[r^\Sigma | \pi_2] < 0$.*

*Proof sketch.* Using $r^{\text{final}} \in \{0, 1\}$, $r^{\max} = r^{\text{final}} + \lambda r^{\text{sub}}(1 - r^{\text{final}})$. With $\mathbb{E}_\pi[\varepsilon \mid r^{\text{final}} = 0] = c$ and $\Pr_\pi(r^{\text{final}} = 0) = 1 - V_f(\pi)$, we get $\mathbb{E}[r^{\max} | \pi] = V_f(\pi) + \lambda c(1 - V_f(\pi)) = (1 - \lambda c)V_f(\pi) + \lambda c$, which is an affine, strictly increasing transform of $V_f$ if $\lambda c < 1$.

To see that ProcessRM-sum does not preserve orders, note that

$$\mathbb{E}[r^\Sigma | \pi_1] - \mathbb{E}[r^\Sigma | \pi_2] = (1 + \lambda\alpha)\underbrace{\left(V_f(\pi_1) - V_f(\pi_2)\right)}_{>0} + \lambda \underbrace{\left(\mathbb{E}_{\pi_1}[\varepsilon] - \mathbb{E}_{\pi_2}[\varepsilon]\right)}_{\text{can be} < -(1+\lambda\alpha)\Delta V_f/\lambda}$$

This theoretical insight is further empirically verified in our experiments.

## 4 EXPERIMENT

We evaluate COGS across multiple artificial image domains to assess its effectiveness in equipping pretrained MLLMs with new reasoning capabilities. We begin with, in Section 4.1, the chart reasoning domain. Using a small subset of questions from the ChartQAPro dataset (Masry et al., 2025a), we show that COGS substantially improves performance on held-out questions. Extending to a mixture of datasets, ChartQAPro + MMC (Liu et al., 2024a), we observe consistent improvements on both datasets, indicating that our framework enables transferable reasoning skills rather than overfitting to a single dataset. Next, in Section 4.2, we evaluate COGS on the VisualWebBench dataset (Liu et al., 2024b), demonstrating that the same approach generalizes beyond the chart domain.

---

*When $r^{\text{final}}$ takes values from $[0, 1]$, ProcessRM-max preserves policy orders iff. $\lambda \cdot r^{\text{sub}} < r^{\text{final}}$.

| Model | Factoid | MCQ | Convers. | FactChk. | Hypoth. | Overall |
|---|---|---|---|---|---|---|
| *Proprietary Models* | | | | | | |
| GPT-5-nano | 45.95 | 63.64 | 49.40 | 63.58 | 49.82 | 50.74 |
| GPT-4o-mini | 43.63 | 66.43 | 45.48 | 59.88 | 45.20 | 48.32 |
| Gemini 2.5 Flash-Lite | 40.42 | 19.96 | 48.77 | 37.43 | 16.66 | 38.72 |
| Claude Haiku 3.5 | 43.44 | 65.03 | 39.84 | 61.79 | 38.77 | 46.74 |
| *Opensource Models (7B+)* | | | | | | |
| Qwen2.5-VL-7B (base) | 42.07 | 62.59 | 44.88 | 60.78 | 50.72 | 47.36 |
| InternVL3.5-GPT-OSS | 43.02 | 58.74 | 42.86 | 58.02 | 54.48 | 46.86 |
| Phi-4-14B | 23.18 | 34.27 | 40.93 | 46.91 | 36.31 | 31.61 |
| *Chart Specialist Models* | | | | | | |
| ChartLLaMA | 8.11 | 23.08 | 18.37 | 45.06 | 29.55 | 17.19 |
| ChartMoE | 19.03 | 35.66 | 32.97 | 45.68 | 27.08 | 27.28 |
| *Prompting Strategies: over Qwen2.5-VL-7B* | | | | | | |
| Self-Consistency | 43.44 | 61.54 | 44.00 | 59.82 | 41.76 | 47.22 |
| Tree of Thoughts | 40.01 | 57.94 | 41.55 | 54.13 | 53.35 | 44.44 |
| Decompositional CoT | 42.08 | 65.03 | 42.57 | 56.53 | 45.55 | 46.36 |
| *Data Synthesis Approaches: over Qwen2.5-VL-7B* | | | | | | |
| ChartQA-Train | 38.77 | 60.14 | 49.72 | 61.11 | 53.12 | 46.64 |
| Chart-R1 | 42.17 | 46.85 | 50.53 | 61.11 | 55.55 | 47.32 |
| In-Context Q Example | 46.33 | 62.94 | 46.91 | 61.11 | **61.72** | 50.58 |
| **COGS (Ours)** | **46.88** | **65.73** | **51.16** | **61.85** | 58.25 | **52.02** |

Table 1: Accuracy (%) on ChartQAPro grouped by question type. COGS performs the best.

Finally, in Section 4.3, we conduct a series of ablation studies to better understand the sources of these improvements. Specifically, we examine: (i) which categories of questions benefit the most, and (ii) the comparative effectiveness of different reward models.

## 4.1 CHART UNDERSTANDING

### 4.1.1 GENERALIZATION FROM SEED TO TARGET DATASET

Chart Question Answering (CQA) requires interpreting visual representations in charts and reasoning over their spatial relation and underlying data. The recently released ChartQAPro benchmark (Masry et al., 2025a) consists of 1,948 human-curated question–answer pairs targeting complex reasoning over diverse chart types. Unlike earlier chart QA datasets that often rely on synthetic or templated questions, ChartQAPro emphasizes natural, high-quality queries that demand multi-step reasoning and interpretation. This makes it a rigorous testbed for evaluating the visual reasoning ability of multimodal language models.

**Setup.** Since ChartQAPro's training and validation sets are not publicly available, we randomly select 33% of the released test set as validation data and treat them as seed questions for data synthesis. The remaining 67% is held out as a fully unseen test set for all experiments. We compare COGS against state-of-the-art pretrained multimodal large language models (MLLMs), chart-specialist models, and recent data synthesis approaches. For all data synthesis methods, including COGS, we use the training set of ChartQA (Masry et al., 2022) as the image source, in order to avoid any contamination from the evaluation data.

**Baselines**. We consider the following models when evaluating COGS:

- **Proprietary Models**: We include representative proprietary models as reference baselines, focusing on small but competitive variants: GPT-5 nano, GPT-4o mini, Gemini 2.5 Flash, and Claude Haiku 3.5.
- **General MLLMs**: We compare against recent open-source general-purpose MLLMs of comparable sizes, including Qwen2.5-VL-7B (Bai et al., 2025), InternVL-3.5-GPT-oss (Wang et al., 2025a), and Phi-4-14B (Abdin et al., 2024).
- **Chart Specialist Models** We consider models specifically designed for chart understanding, including ChartLLaMA (Han et al., 2023), which improves chart QA performance after training

on high-quality synthetic instruction data, and ChartMoE (Xu et al., 2025) which integrates a Mixture-of-Experts (MoE) architecture to facilitate chart understanding.

- **Prompting Strategies** We compare against 3 prompting strategies at inference time, including self-consistency (Wang et al., 2023b) and Tree of Thoughts prompting (Yao et al., 2023). We also introduce inference-time decomposition as an additional variants of our method. In this setting, the LLM is prompted at inference time to decompose a complex question into simpler perception and reasoning subquestions, using the same decomposition instructions as in our factor pool construction.

- **Data Synthesis Approaches.** We compare COGS against other data synthesis methods including (1) the original QA pairs in ChartQA training set which is generated by machine, (2) ChartR1 (Chen et al., 2025) which programmatically synthesizes chart reasoning data and conduct reinforcement finetuning; and (3) In-Context Question Examples where we follow question synthesis convention to generate questions with in-context question examples. Example questions are sampled from the seed dataset. Notably, all baselines are fine-tuned with GRPO using the same base model, Qwen2.5-VL-7B (Bai et al., 2025), and image source (the training set of ChartQA) for same training effort, to enable fair comparisons. We used StandardRM due to the absence of subquestions and corresponding answer in these datasets.

**Result.** Table 1 shows the performance on different question types on ChartQAPro. Among open-source MLLMs, Qwen2.5-VL-7B achieves the strongest overall accuracy (47.36%). Proprietary models such as GPT models and Haiku 3.5 perform reasonably well, but remain slightly below the our fine-tuned Qwen2.5-VL-7B using the COGS framework. Chart-specialist models, while tailored to chart understanding, perform poorly compared to COGS. This is largely because they are typically constrained by specially designed architecture and trained on relatively narrow datasets, which do not fully cover the distribution of ChartQAPro and therefore suffer from domain gaps.

All data synthesis approaches demonstrate minor benefits over the base model likely due to domain gaps as well. COGS achieves the highest overall accuracy of 52.02%, outperforming both baselines and all open-source MLLMs by a significant margin. We provide in-depth analysis of the performance gains in Section 4.3.

We observe a substantial performance gap between inference-time decomposition and COGS, largely due to error accumulation across sub-questions. Instead, COGS mitigates this issue by rewarding correct intermediate substitutions in training, reducing error compounding. Moreover, RL training in COGS enables the model to flexibly integrate decomposition signals without being constrained to a single reasoning path, unlike inference-time decomposition where the provided examples in the context may restrict the model's ability to explore novel reasoning paths.

### 4.1.2 GENERALIZATION OVER MIXTURE OF DATASETS

**Setup.** We extend COGS to a multi-dataset setting by incorporating the MultiModal Chart Benchmark (MMC-Bench) (Liu et al., 2024a), a recent CQA dataset with reasoning-intensive, human-annotated QA pairs. Similar to our ChartQAPro (noted as *seed A*) setup, we split the MMC-Bench test set into 33% validation questions (used as *seed B*) and 67% held-out test questions.

**Variants.** We compare two strategies for synthesizing data across domains: 1. **Data-level mixture**: decompose and recompose A and B independently, then combine the synthesized data, i.e., $Recompose(Decompose(A)) + Recompose(Decompose(B))$. 2. **Factor-level mixture**: decompose A and B separately, merge all extracted factors into a joint pool, and recompose using this combined pool, i.e., $Recompose(Decompose(A) \cup Decompose(B))$. In addition, we include two "specialist models" trained only with augmented data from a single domain (e.g., trained on augmented A and evaluated on A). These serve as "upper-bound references" for in-domain data augmentation. All methods use Qwen2.5-VL-7B as the base model and are trained with GRPO and ProcessRM-max.

**COGS shows transferrable benefits across datasets.** As shown in Table 2, both data-level and factor-level mixtures substantially improve performance in both domains, demonstrating that COGS facilitates positive transfer across datasets rather than simply overfitting to one. Crucially, the factor-level mixture consistently outperforms the data-level mixture, suggesting that factor recomposition better captures shared structures between domains.

| Model | ChartQAPro | MMC |
|---|---|---|
| Qwen2.5VL | 47.36 | 85.65 |
| + ChartQAPro | **52.02** | 85.69 |
| + MMC | 49.93 | **88.10** |
| + Data-level Mix | 50.72 | 86.99 |
| + Factor-level Mix | **52.33** | 87.55 |

Table 2: Multi-data co-training results.

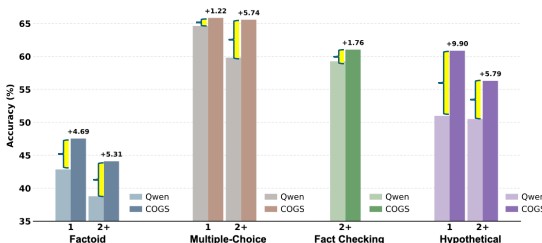 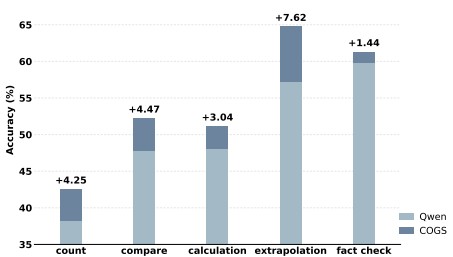

Figure 3: Accuracy (%) on ChartQAPro by reasoning factor numbers and question types. COGS generally yields the larger gains on questions with more factors.

Figure 4: Accuracy (%) on ChartQAPro by reasoning factor types with complexity from low to high.

**Factor-level mixture is a better strategy for data mixing.** We observe that factor-level mixture consistently outperforms data-level mixture, and achieves performance on both domains comparable to specialist models trained exclusively in-domain. This suggests that factorization offers a more effective way to leverage multiple datasets. Prior research on *data mixing* and *multi-dataset training* has shown that simply combining heterogeneous datasets often fails to yield the best generalization, as models may overfit to dominant distributions or under-utilize complementary signals. By breaking down questions into primitive factors before recomposition, COGS provides a common representational ground across datasets, enabling more transferable training. This suggests a promising direction for the long-standing challenge of data mixture in foundation model training.

## 4.2 WEBPAGE GUI UNDERSTANDING

To demonstrate the generality of COGS, we also evaluate it on the webpage question answering domain, which requires visual, semantic, and structural reasoning over graphical user interfaces (GUIs). We adopt VisualWebBench (Liu et al., 2024b), a benchmark consisting of diverse real-world webpages paired with reasoning-intensive, human-curated questions. We use questions from VisualWebBench as seeds and screenshots from MultiUI (Liu et al., 2025) as the image source.

**Setup.** We evaluate the same set of proprietary and general-purpose MLLMs as in the chart understanding experiments. In addition, we compare against the GUI specialist UIX-Qwen2 and data synthesis approach in Liu et al. (2025). We sampled 10k webpage QA data from MultiUI (Liu et al., 2025), and fine-tuned Qwen2.5-VL-7B (Bai et al., 2025) with GRPO (Shao et al., 2024). The results are reported as MultiUI-WQA in Table 3.

**Result.** Table 3 shows the result. Qwen2.5-VL-7B achieves 85.65% accuracy, outperforming most open-source baselines, while specialist models such as UiX-Qwen2 lag behind. Inference-time decomposition yields minor gain (86.12%). Among these, COGS achieves the best non-proprietary result at 88.04%. These findings confirm that COGS generalizes beyond charts, effectively boosting reasoning capability over complex webpages.

## 4.3 ADDITIONAL ANALYSIS

In this section, we attribute the reasoning capability gains of COGS to two factors: (1) enhanced performance on reasoning-intensive questions, including multi-hop reasoning (Fig. 3) and complex reasoning factors (Fig. 4), and (2) the impact of different reward models.

| Model | WebQA |
|---|---|
| ***Proprietary Models*** | |
| GPT-5-nano | 89.47 |
| GPT-4o-mini | 81.34 |
| Gemini 2.5 Flash-Lite | 81.85 |
| Claude Haiku 3.5 | 80.86 |
| ***Opensource Models (∼7B)*** | |
| Qwen2.5-VL-7B (base model) | 85.65 |
| InternVL3.5-GPT-OSS | 74.64 |
| Phi-4-14B | 74.16 |
| ***Specialist Models*** | |
| UiX-Qwen2 | 68.90 |
| ***Inference-time decomposition*** | |
| Decompositional CoT | 86.12 |
| ***Data Synthesis Approaches*** | |
| MultiUI-WQA | 86.60 |
| COGS (Ours) | **88.04** |

Table 3: Accuracy (%) on Visual-WebBench. COGS performs the best among all non-proprietary models.

**COGS improves multi-hop reasoning.** Fig. 3 shows model performance grouped by the number of factors. Overall, the performance improvement becomes more pronounced as questions have longer reasoning chains. This trend holds across factoid, multiple-choice, and fact-checking questions. For hypothetical questions, however, the trend is less salient: we conjecture that their difficulty is already

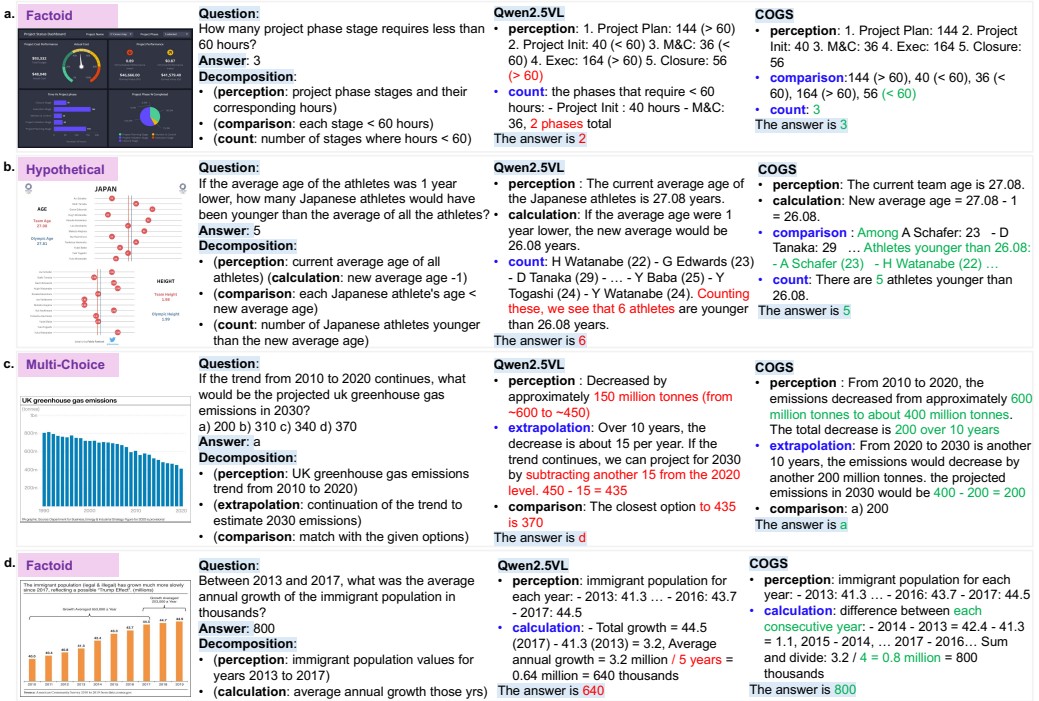

Figure 5: Qualitative evaluation examples. COGS-RL improved base models on questions that contain multiple factors and from different question types in ChartQAPro.

dominated by the first counterfactual reasoning factor (*"if xxx happens, ..."*). Therefore, adding more factors does not compound the hardness in the same way.

This observation is further illustrated by the substantial gains on the *Count* (+4.25%) and *Compare* (+4.47%) factors in Figure 4. These two factors frequently co-occur as essential steps in multi-hop reasoning, such as *Counting* values based on results of *Comparison* operations. As illustrated in Figure 5 (Rows **a** and **b**), models trained with COGS better capture such compositional structures, whereas baseline models tend to shortcut the process and directly produces the answer (which can be flawed, e.g., being number-insensitive and incorrectly concluding $56 > 60$, as shown in Row **a**).

**COGS supports advanced reasoning factors.** At the factor level, we also observe strong gains on advanced reasoning factors such as *Extrapolation* (+7.62%) and *Calculation* (+3.04%) in Figure 4. These factors require models not only to execute operations but also to decide which operations are appropriate (e.g., whether to add, divide, or apply another function). The complexity is illustrated in Figure 5 (Rows **c** and **d**). By training models on diverse reasoning trajectories with our compositional data generation framework, we can improve factor-level reasoning performances. For example, in Row **d**, COGS correctly identifies that the average annual growth should be computed over 4 intervals, rather than mistakenly dividing the difference between the first and last year by 5.

**Ablation: reward model.** We conduct an ablation study on the reward models under three settings proposed in Section 3.4, with GRPO and COGS data seeded from ChartQAPro. As shown in Table 4: ProcessRM-sum slightly worsens performance, while ProcessRM-max consistently improves it compared to StandardRM. This is consistent with our theoretical analysis in Proposition 3.1, which shows that ProcessRM-max preserves policy order under noisy sub-reward signals, whereas ProcessRM-sum does not.

| Reward Model | Overall Acc. |
|---|---|
| StandardRM | 50.96 |
| ProcessRM-sum | 50.35 |
| ProcessRM-max | **52.02** |
| **Alt. Training Setting** | |
| SFT+ProcessRM-max | 46.62 |

Table 4: Ablation study on reward models shows that ProcessRM-max maximally boosts the model performance.

We further ablate the training strategy by an additional supervised fine-tuning (SFT) phase with 35k COGS examples prior to GRPO. To mitigate overfitting,

the datasets for SFT and GRPO are non-overlapping. Results show that while SFT helps regulate output format, it does not enhance reasoning ability, consistent with findings from Chu et al. (2025).

**Ablation: size of seed questions.** We also ablate the seed set size used for COGS data synthesis on ChartQAPro. We held out 67% of ChartQAPro as a fixed evaluation set for fair comparison and sampled 1%, 5%, 15%, 25%, and 33% of the original data size from the remaining 33%. We then trained Qwen2.5-VL-7B on data generated from each seed set. As shown in Fig. 6, performance increases with seed size. Questions generated from very small seed sets are less representative, resulting in relatively poor performance, whereas a reasonable subset such as 33% already yields a substantial boost.

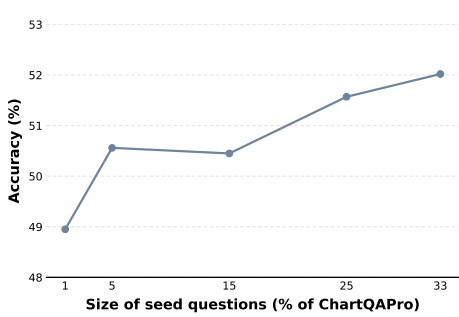

Figure 6: Ablation study on the size of seed questions for COGS on ChartQAPro. Performance improves as the size grows.

## 5 CONCLUSION

We have introduced COGS, a data-efficient framework for equipping pretrained multi-modal large language models with new reasoning capabilities in domains where annotated question–answer data is scarce. The key idea is to decompose seed questions into primitive factors, and then systematically recompose these factors with new images to generate diverse, compositional training data.

**Future Work.** This work opens several future directions. First, our experiments focus on single charts and single webpage screenshots; extending COGS to long-context reasoning over visually rich documents will broaden its scope. Second, it is important to study how our data synthesis can be integrated into the pretraining stage of MLLMs or combined with search algorithms to further boost process reward guidance (Zhang et al., 2024a; Park et al., 2025). Third, future work may consider investigating how the reasoning capabilities acquired through COGS transfer to downstream tasks—such as chart code editing or web agent applications.

## ETHICS STATEMENT

The primary contribution of this work is an efficient data augmentation pipeline that factorizes a small set of seed data into diverse reasoning question–answer pairs. All data used in our work that are already publicly released and open-sourced under their respective licenses, which we carefully followed. Our method does not generate new images or introduce additional modalities. We make sure that the synthesized question–answer pairs focus only on reasoning over charts and web pages, avoiding offensive, biased, or sensitive content. As such, the ethical considerations remain consistent with those already established for the underlying datasets and models. We hope we further promote reproducibility and transparency through the release of code and augmented data.

## REPRODUCIBILITY STATEMENT

To ensure transparency, we rely solely on publicly available resources: all open-source LLM weights are downloaded from their official repositories, and proprietary models are accessed via their documented code and APIs. We describe all experimental configurations, including prompt templates, hyperparameter choices, and software/hardware environments, in Section 3, Section 4, and the Appendix D. Our experiments do not involve any private or sensitive data. We release the code and data at https://cogsynthesis.github.io.

## ACKNOWLEDGMENTS

We thank Lei Xu, Bowen Pan, Pingchuan Ma and members of MIT-IBM Watson AI Lab for helpful discussions and valuable feedback.

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

APPENDIX

## A   THE USE OF LARGE LANGUAGE MODELS (LLMS)

We use ChatGPT as a grammar checker for the writing of this paper. We also use small proprietary language models as evaluation baselines to compare performance in our experiments as described in Section 4. We use open-sourced MLLM in our synthetic data generation pipeline follow corresponding license.

## B   QUALITATIVE EVALUATION ON VISUALWEBBENCH

We provide some visualization examples of evaluation on VisualWebBench. COGS has been largely improved in questions involving reasoning like comparison, as well as spatial relation.

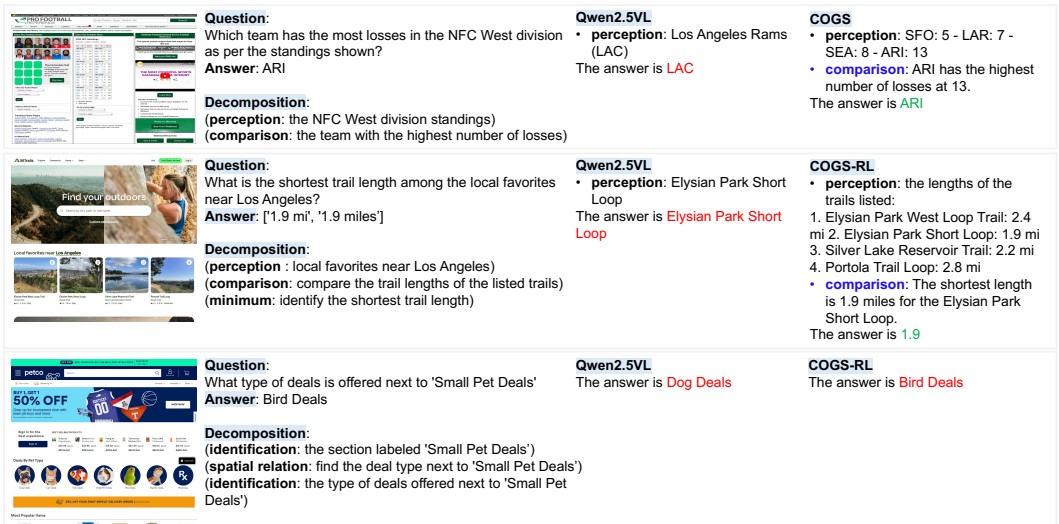

Figure 7: Example of evaluation on VisualWebBench.

## C   ABLATION STUDY ON BASE MODELS

We evaluate whether the gains from COGS depend on model capacity within a family or on the choice of model family at a fixed parameter scale. Concretely, we fine-tune Qwen2.5-VL-3B (same family as our main experiments but smaller size) and LLaVA-1.5-7B (different family at 7B scale) using the same COGS dataset and the same RL configuration as in the main experiments. All training hyperparameters, reward settings, and evaluation protocols are kept identical to isolate the effect of the base model. We then evaluate on the fixed ChartQAPro test split using the same metrics reported in the main table.

Across both comparisons, COGS consistently improves the corresponding base model. These results indicate that the benefit of COGS is not specific to a particular parameter count or to a single model family, and that the decomposition-guided synthesis remains effective under changes in backbone capacity and architecture. Full results are reported in Table 5.

| Model | Overall (Base) | Overall (+COGS) |
|---|---|---|
| Qwen2.5-VL-3B | 36.22 | **38.68** |
| LLaVA-v1.5-7B | 13.47 | **22.04** |

Table 5: Accuracy (%) on ChartQAPro comparing base models and +COGS.

## D   MORE DETAILS ABOUT COGS IMPLEMENTATION AND REPRODUCIBILITY

**Hyper-parameters**   We use verl(Sheng et al., 2025) for GRPO training. We ran with epoch = 4 using a large-scale distributed setup with 8 GPUs per node across 4 nodes. The model was trained with batch size = 1024 and maximum input/output lengths = 4096 tokens for prompts and 2048 tokens for responses, respectively. Optimization used a learning rate = 1e-6, with GRPO updates performed on mini-batches of 256. To stabilize training, we applied a $KL$-penalty loss with a coefficient = 0.001, while disabling $KL$ in the reward. Gradient checkpointing was enabled for memory efficiency, and tensor model parallelism was set to size 2. Rollouts used 16 samples per step, with GPU memory utilization capped at 0.6.

**Running Software/Hardware Environment and Training Time**   Our implementation is based on Python, with Transformers v4.51.3, PyTorch v2.6.0, and CUDA 12.4. We use VERL v0.5.0.dev0 to fine-tune models with Reinforcement Learning via GRPO Shao et al. (2024). All experiments are distributed across 4 nodes, each equipped with 8 NVIDIA H100 GPUs (80GB). Training data is generated using Qwen2.5-vl-72B. Fine-tuning the base Qwen2.5-vl-7B model takes approximately 10 hours with GRPO and 2 hours with SFT on the Chart Understanding task with 10k reasoning examples, while the WebGUI QA task (also 10k examples, but web images are larger) requires about 21 hours of GRPO.

**Evaluation**   We sampled 67% from each benchmark dataset for evaluation. For evaluation on chart question answering benchmarks, We adopted official prompt templates for each question category under the chain of thought setup released in original paper. For webpage GUI question answering, we enable the chain of thought by following prompt: You will be given an image and a question that you need to answer based on the provided image. You need to think step-by-step and format the final answer in a separate sentence like "The answer is X". The final answer should be in the fewest words possible. We use lmms-eval for evaluation(Zhang et al., 2024b; Li et al., 2024a).

**Prompt for Seed Question Decomposition and New Question Recomposition are included in the next section.**

# E  PROMPTS

This section specifies the prompt templates we use to decompose questions from seed dataset for the factor pool and re-composition of questions for COGS dataset.

## E.1  PROMPTS FOR QUESTION DECOMPOSITION

**Decomposition Prompt.**

> We can decompose each question into subquestions from one of the general types. Here are some examples:
> [in-context example: Chart/Web]
>
> Please do the same for the following questions in the same format without explanation.
> Check the information in the attached image carefully. If the question can be easily answered with a simple identification step, avoid unnecessary decomposition.
> Remember to strictly follow the format of the example, and don't provide the answer.
> <image>
> Question: **{query}**

**In-context question-decomposition example: Chart** for both ChartQAPro and MMC in this paper

> (Question: How many times has the satisfied rate been above 25%?) = (identification: satisfied rate of each year) + (comparison: each instance $> 25\%$) + (count: number of instances where satisfied rate $> 25\%$)
> (Question: Is the following statement True or False? Gen X has experienced a steeper population increase than baby boomers did between 1990 and 2015.) = (identification: Gen X's population increase curve) + (identification: baby boomers' increase curve) + (comparison: which one has a steeper curve) + (fact checking: given the finding from the previous step, is the statement true?)
> (Question: if a multi-college district served 10,000 students, how many students were determined eligible using EFC criteria?) = (identification: percentage of students determined eligible using EFC criteria in a multi-college district) + (calculation: number of students based on that percentage)
> (Question: if the actual Avg ACA Premium in 2017 had turned out to be $5,000, and the +30% label accurately reflected the difference compared to the Low Est. projection for that hypothetical $5,000 value, what would be the implied Avg Individual Mrkt Premium Without ACA - Low Est. - in 2017?) = (identification: Avg ACA Premium in 2017) + (identification: +30% label that reflects the difference of Avg ACA Premium in 2017 compared to Low Est.) + (calculation: implied Low Est. value based on the given 30% difference and hypothetical $5,000 ACA Premium)

**In-context question-decomposition example: Webpage GUI** for VisualWebBench in this paper

> (Question: How many times has the satisfied rate been above 25%?) = (identification: satisfied rate of each year) + (comparison: each instance $> 25\%$) + (count: number of instances where satisfied rate $> 25\%$)
> (Question: Is the following statement yes or no? Gen X has experienced a steeper population increase than baby boomers did between 1990 and 2015.) = (identification: Gen X's population increase curve) + (identification: baby boomers' increase curve) + (comparison: which one has a steeper curve) + (fact checking: given the finding from the previous step, decide yes or no)
> (Question: According to this chart, what is the revenue of Retailer D at Month 6?) = (identification: revenue of Retailer D at Month 6)

## E.2 PROMPTS FOR QUESTION RE-COMPOSITION

Given the following chart:
chart: <image>
Your Task is to generate 5 sets of question-answer pairs for instruction tuning. In each set of QA pairs, you need to first identify **{perception_count}** entities, and then compose **{reasoning_count}** level of reasoning questions related to them. The 2-nd order reasoning questions should be based on the answers of the 1-st order reasoning questions, and so on. Each question must meet ALL these conditions:
1. Content Source: Only use data present in the given chart.
2. Structure: Each question must include exactly **{count1} {factor1}**, ... , and **{countN} {factorN}**. Each identification question should ask about one and only one entity/concept, the following **{reasoning_factors}** subquestions should be the question related and only related to the entities/concept mentioned in the previous subquestions. The specific example of each subquestion type will be provided in the following text.
3. Content: Each question must be based on the chart data, and can be answered using natural language. Avoid asking about the size of an object that is not relevant to the data (e.g., font size of a label).
4. Relevance: If there is a reasoning subquestion, it must operate on the entities or values identified in the observation subquestion. [in-context example 1]
5. Conciseness: After writing the detailed question, provide a natural concise version. This concise version should still look like a question, and can be asked independently without the previous question. [in-context example 2]
6. Answer: Provide a step-by-step reasoning for how you found the answer.
7. Final Answer: Provide just the concise final answer to the concise question, without any explanation or reasoning.
Reference Examples: **{factors} {sampled_subquestion_of_the_factor}**
Here are some examples of the concise questions: **{sampled_concise_questions}**

Expected Output Format for the generated questions: Use the following structure for each pair:
[in-context example 3]

Instructions:
1. Follow the example strictly. If the question contains reasoning subquestions, make sure it is relevant to the observation questions.
2. Use only the given data in the chart.
3. Provide exactly 5 unique Q&A pairs. [question_types]
4. Validate each answer. The answer must be grounded to the data shown in the chart.
5. Each pair must include both detailed step-by-step reasoning and the final result.

Generate Now:
Please proceed with generating your 5 question-answer pairs now.

**In-Context Example 1**

For example, if the observation subquestion asked about the value of A and B, and you are asked to generate a calculation subquestion after them, it must be some calculation between A and B. If there are multiple levels of reasoning questions, the later reasoning subquestions should be based on the answers of the previous subquestions. Do not ask irrelevant questions. For example, if the first subquestion is "what's the difference between A and B", an acceptable next-level reasoning question would be "what's the difference between A and B compared to C". You should avoid an unacceptable question like "what's the difference between A and B and what's the difference between A and C".

**In-Context Example 2**

For example, if the detailed question asks about a new value if A is changed, the concise question cannot simply refer to a "new value" without mentioning it depends on A being changed. An explicit example: detailed question: "What is A, what is B, what is the new value of A+B if A is changed to 10, what is the difference between the new value and C?" A bad concise question is: "What is the difference between new value and C?", because it does not mention A is changed to 10, and it does not mention A+B is the new value. A correct concise question should be: "What is the difference between the new value of A+B and C, if A is changed to 10?"

**In-Context Example 3**

```
{
  1: {
    "Question": "<Full question with two identifications and one
        comparison>",
    "Concise question": "<Concise version of the question>",
    "Answer": "<Step-by-step reasoning and calculation>",
    "Final Answer": "<The final answer to the concise question>"
  },
  2: {
    "Question": "...",
    "Concise question": "...",
    "Answer": "...",
    "Final Answer": "..."
  },
  ...
}

Example (not actual data):
{
  1: {
    "Question": "What was the percentage for Technology, what was the
        percentage for Finance, and what is the difference between
        them?",
    "Concise question": "What is the difference between Technology
        and Finance's percentages?",
    "Answer": "Step 1: Technology's percentage is 23.7%. Step 2:
        Finance's percentage is 26.3%. Step 3: The difference is
        |23.7% - 26.3%| = 2.6%.",
    "Final Answer": "2.6%"
  }
}
```

# F   COGS DATA EXAMPLE

## F.1   VISUALIZATION OF SELECTED COGS-CHARTQAPRO

We provide visualization examples of COGS-ChartQAPro Datasets.

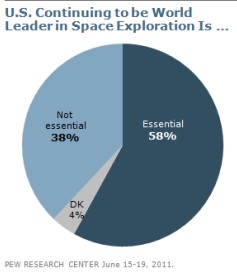

**Complex Question**:
Which is the correct answer to the following question: if the percentage of 'DK' respondents was added to 'Not essential', what would be the new total percentage of 'Not essential'?
a) 42% b) 44% c) 46% d) 48%
**Answer:**
a) 42%

**Sub-Questions**:

Q1: What is the percentage of respondents who responded 'DK'?
A1:4

Q2: What is the percentage of respondents who think it is not essential?
A2: 38

Q3: The percentage of 'DK' respondents is 4%, and the percentage of those who think it is not essential is 38%. What would be the new total percentage of 'Not essential' if the percentage of 'DK' respondents was added to 'Not essential'?
A3: a) 42%

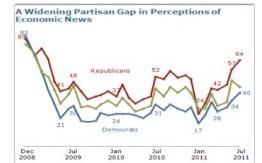

**Complex Question**:
Is the following statement True or False: the score of Republicans in Dec 2008 is higher than the score of Democrats in Jan 2010?
**Answer:**
True

**Sub-Questions**:

Q1: What was the perception score of Republicans in Dec 2008?
A1: 82

Q2: What was the perception score of Democrats in Jan 2010?
A2: 24

Q3: Is the score of Republicans in Dec 2008 higher than the score of Democrats in Jan 2010?
A3: True

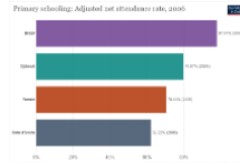

**Complex Question**:
Which country has a higher adjusted net attendance rate in 2006, Brazil or Djibouti?
**Answer:**
Brazil

**Sub-Questions**:

Q1: What was the adjusted net attendance rate for Brazil in 2006?
A1: 97.91

Q1: What was the adjusted net attendance rate for Djibouti in 2006?
A2: 79.87

Q3: Which country has a higher adjusted net attendance rate, Brazil or Djibouti?
A3: Brazil

Figure 8: Example of complex questions with 3 subquestions.

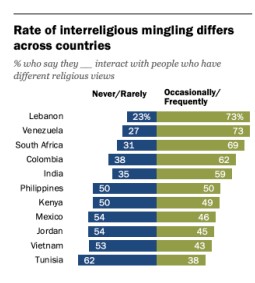

**Complex Question**:
Is the following statement True or False? The result of subtracting the swapped 'Never/Rarely' percentage from the swapped 'Occasionally/Frequency' percentage is positive for South Africa.
**Answer:**
False

**Sub-Questions**:

Q1: What is the percentage of people who interact 'Never/Rarely' in South Africa?
A1: 31

Q2: What is the percentage of people who interact 'Occasionally/Frequently' in South Africa?
A2: 69

Q3: Swap the two percentages identified. The percentage of people who interact 'Never/Rarely' in South Africa becomes 69%, and the percentage of people who interact 'Occasionally/Frequently' in South Africa becomes 31%. What is the result of subtracting the swapped 'Never/Rarely' percentage from the swapped 'Occasionally/Frequency' percentage?
A: -38

Q4: Is the following statement True or False? The result of subtracting the swapped 'Never/Rarely' percentage from the swapped 'Occasionally/Frequency' percentage is positive for South Africa.
A4: False

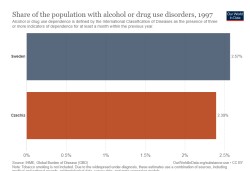

**Complex Question**:
What is the new sum of Sweden and Czechia's shares of population with alcohol or drug use disorders, if the share in Sweden decreases by 10%?
**Answer:**
4.703

**Sub-Questions**:

Q1: What is the share of the population with alcohol or drug use disorders in Sweden?
A1: 2.57

Q2: What is the share of the population with alcohol or drug use disorders in Czechia?
A2: 2.39

Q3: If the share in Sweden decreases by 10%, what is the new share in Sweden?
A3: 2.313

Q4: What is the new sum of Sweden and Czechia's shares of population with alcohol or drug use disorders, if the share in Sweden decreases by 10%?
A4: 4.703

**Complex Question**:
Does the adjusted value of Google's share surpass Facebook's share after a 5% increase?
**Answer:**
False

**Sub-Questions**:

Q1: What is the share of mobile display ad revenues for Facebook?
A1: 35.7

Q2: What is the share of mobile display ad revenues for Google?
A2: 15.4

Q3: What is the difference between the shares of mobile display ad revenues for Facebook and Google?
A3: 20.3

Q4: Apply a 5% increase in Google's share and check if the adjusted value surpasses Facebook's share
A4: The adjusted value of Google's share is 16.17, which does not surpass Facebook's share of 35.7

Figure 9: Example of complex questions with 4 subquestions.

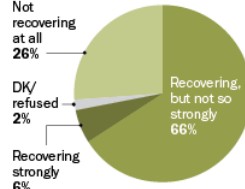

**Feeling the Recovery Blahs**

*Which best describes your opinion: The economy is...*

Not recovering at all 26%

DK/ refused 2%

Recovering strongly 6%

Recovering, but not so strongly 66%

Source: Pew Research Center/USA TODAY survey conducted April 23-27, 2014

PEW RESEARCH CENTER

**Complex Question**:
Is the combined percentage of 'Not recovering at all' and 'Recovering strongly' more than double the 'DK/refused' percentage?
**Answer:**
True

**Sub-Questions**:

Q1: What is the percentage of people who think the economy is 'Not recovering at all'?
A1: 26

Q2: What is the percentage of people who think the economy is 'Recovering strongly'?
A2: 6

Q3: The combined percentage of those who think the economy is either 'Not recovering at all' or 'Recovering strongly'?
A3: 32

Q4: What is the percentage of people who think the economy is 'DK/refused' doubled?
A4: 4

Q5: Is the combined percentage of 'Not recovering at all' and 'Recovering strongly' more than double the 'DK/refused' percentage?
A5: True

**Complex Question**:
Is the difference between Tertiary and Primary GPI greater or smaller than the difference between Primary and Secondary GPI?
**Answer:**
Greater

**Sub-Questions**:

Q1: What is the GPI for Tertiary education?
A1: 1.01

Q2: What is the GPI for Primary education?
A2: 0.98

Q3: What is the GPI for Secondary education?
A3: 0.98

Q4: The GPI for Tertiary education is 1.01 and the GPI for Primary education is 0.98. What is the difference between these two values?
A4: 0.03

Q5: The GPI for Primary education is 0.98 and the GPI for Secondary education is 0.98. What is the difference between these two values
A5: 0

Q6: The difference between Tertiary and Primary GPI is 0.03, and the difference between Primary and Secondary GPI is 0. Is the difference between Tertiary and Primary GPI greater or smaller than the difference between Primary and Secondary GPI?
A: Greater

Figure 10: Example of complex questions with more than 4 subquestions.

## F.2 Visualization of selected COGS-VisualWebBench

We provide visualization examples of COGS-VisualWebBench Datasets.

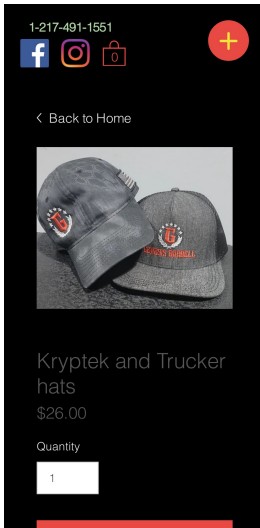

**Complex Question**:
What would be the total cost of Kryptek and Trucker hats if the quantity is increased by 2?
**Answer:**
78

**Subquestions**:

Q1: What is the price of the Kryptek and Trucker hats?
A1: 26

Q2: Calculate the total cost if the quantity is increased by 2?
A2: 78

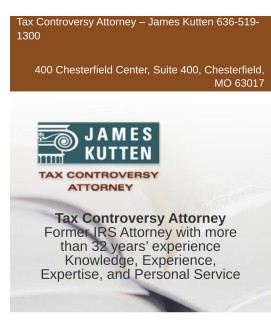

**Complex Question**:
What is the full zip code of the city where James Kutten's office is located?
**Answer:**
63017

**Subquestions**:

Q1: Find the city next to the address.
A1: Chesterfield

Q1: What is the full zip code of this city?
A2: 63017

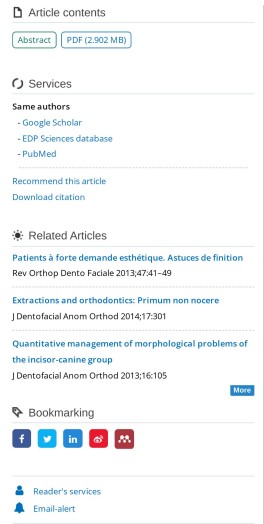

**Complex Question**:
What is the difference in years between the publication dates of the first and third articles listed under Related Articles?
**Answer:**
0

**Subquestions**:

Q1: What is the publication dates of the first articles listed under Related Articles?
A1: 2013

Q2: What is the publication dates of the third articles listed under Related Articles?
A2: 2013

Q3: What is the difference
A3: 0

Figure 11: Example of complex questions seeded from 33% VisualWebBench.

