# OpenReview forum: "Composition-Grounded Data Synthesis for Visual Reasoning"
_ICLR.cc/2026/Conference — ICLR 2026 Poster_

### Official Review · Reviewer_dvz5 · 2025-10-18

**Soundness:** 2
**Presentation:** 2
**Contribution:** 2
**Rating:** 4
**Confidence:** 4

**Summary:**

The paper proposes COGS (COmposition-Grounded instruction Synthesis), a three-stage pipeline to improve VLMs’ reasoning in artificial image domains (charts, rendered documents, webpages). Stage 1 decomposes a seed set of questions into interpretable perception and reasoning factors; Stage 2 recomposes sampled factors with new images to synthesize new questions; Stage 3 fine-tunes VLMs using GRPO with the generated questions. Experiment demonstrates the effectiveness of COGS on ChartQAPro and MMC-Bench.

**Strengths:**

1.	The pipeline of COGS is clear and easy to follow. The method is intuitive.

2.	Factor-level sub-questions make the supervision more transparent and are practical for error analysis.

**Weaknesses:**

1.	Evaluation protocol risks leakage. The use of 33% of the test set as seeds violates the test-only usage and can tune the pipeline to the test distribution even without answer leakage. I checked the ChartQAPro benchmark, and it has 1341 charts for 1948 questions. Different questions may target the same image chart; this makes image-level leakage likely.

2.	The process reward uses LLM-as-a-judge, which can cause significant training overhead. The paper should report efficiency statistics

**Questions:**

Please see Weaknesses

---

> ### Author Response · Authors · 2025-11-22
>
> **W1. Image from the seed data.** We designed the pipeline to avoid both answer and image-level leakage.
>
> 1. Source separation. We use ChartQAPro only for decomposition to mine text-level factors. No ChartQAPro images are ever used in the recomposition step.
> 2. Disjoint images. All synthetic questions are grounded on images drawn exclusively from the ChartQA training split, which is fully disjoint from ChartQAPro. Thus, image-level leakage from ChartQAPro is impossible—none of its images enter our training data.
> 3. No reuse of test content. During recomposition, numeric values and textual references are bound to the new ChartQA-train images. We do not use any ChartQAPro content in this step
>
> Regarding the concern that using a portion of ChartQAPro as “seeds” might tune the model to its test distribution: we emphasize that seed questions are sampled from a disjoint 33% subset of ChartQAPro, while the remaining 67% is reserved exclusively for testing. Since we extract only abstract factors from the seed subset, and all grounding occurs on a completely separate image pool, the model is never exposed to ChartQAPro’s images or the specific QA content used for evaluation.
>
> **W2. Computation analysis.** LLM-as-a-judge (or LLM-based reward modeling) is now a common choice for reward scoring and for training LLM-based reward models, as demonstrated in widely cited works [1–4]. In contrast to methods that rely on multiple judge queries or long free-form critiques, our design keeps each reward call short and efficient. In COGS, each step includes a reference sub-question and reference answer, so the judge returns only a scalar match score rather than a long explanation. This keeps prompts compact and significantly reduces inference cost.
>
> We thank the reviewer for suggesting an explicit evaluation of efficiency. Below, we report measurements under a controlled setup of 4 GPUs for model training and 4 GPUs for the reward service. According to implementation logs:
>
> - 34% of total time is spent on actor updates (backpropagation + optimizer),
> - 39% on old-logprobs forward passes, reference-model logprobs, and actor rollouts, and
> - 26% on reward inference.
>
> Thus, the reward stage is not the bottleneck. Since our reward inputs are short, pure text and the judge is served through vLLM, we did not observe significant overhead from reward computation in practice.
>
> ---
>
> We again thank the reviewer for the constructive feedback and insightful questions. We hope our clarifications on evaluation setup and experiments on computation analysis have addressed your concerns and helped illustrate the rationale behind our framework design. We are happy to provide any additional details if needed and kindly hope you to consider revisiting your score in light of these clarifications.
>
>
> References
>
> [1] Self-Rewarding Language Models
>
> [2] SALMON: Self-Alignment with Instructable Reward Models
>
> [3] RLAIF vs. RLHF: Scaling Reinforcement Learning from Human Feedback with AI Feedback
>
> [4] Process-based Self-Rewarding Language Models
>
> [5] J1: Incentivizing Thinking in LLM-as-a-Judge via Reinforcement Learning

---

> ### Author Response · Authors · 2025-11-26
>
> Dear reviewer dvz5,
> Thank you for your constructive review. We have added clarifications and experiments that directly address your comments. Please let us know if we can provide anything further. Thank you again for your time and effort, and enjoy the Thanksgiving weekend.

---

### Official Review · Reviewer_jHKn · 2025-10-28

**Soundness:** 2
**Presentation:** 3
**Contribution:** 2
**Rating:** 4
**Confidence:** 4

**Summary:**

This paper proposes COGS (Composition-Grounded Instruction Synthesis), a data-efficient pipeline to endow MLLMs with visual reasoning skills in artificial image domains like charts, documents, webpages. The method (i) decomposes a small set of seed questions into atomic perception/reasoning factors, (ii) recomposes sampled factors with new images to synthesize QA pairs with sub-questions and sub-answers, and (iii) fine-tunes an MLLM via GRPO using process rewards computed at the factor level.  Experiments on ChartQAPro and VisualWebBench show consistent gains over strong open baselines, ablations on mixing factor pools across datasets, reward model, factor break down further demonstrate the effectiveness and scalability of the method.

**Strengths:**

1. Clear, modular idea with practical upside. Factorizing seed questions then reusing those factors across unlabeled images is straightforward and domain-agnostic. The pipeline is well-motivated and broadly applicable.
2. Process reward design with both theoretical and experimental justification. The ProcessRM-max objective is motivated by a simple but persuasive order-preservation argument and turns out to be effective.
3. Transfer via data-level and factor-level mixing. The author provides a valid ablation on generalization over mixture of datasets, indicating the method captures reusable structure, not just surface patterns.

**Weaknesses:**

**1. Positioning in prior data-synthesis work is thin.**

 The introduction and related work don’t really situate the paper within the broader line of data-synthesis methods that first generate sub-questions/functions and then recombine them into new examples. It would help to spell out what’s borrowed vs. what’s new, why existing approaches fall short for this setting, and how the paper’s factorization addresses those limits. Please cite a few representative strands to make the attribution explicit. See [1-3].

**2. Comparisons skip broader synthesis and RL frameworks.**

 The experiments compare against chart/GUI-specific generators, but there are notable data-synthesis pipelines in RL and process-supervised training that should be part of the picture. Even a small, controlled head-to-head with one or two representative frameworks (same base model, same budget) would clarify what the proposed method adds beyond existing synthesis strategies. Some works can be found in [4-5].

**3. Reliability of synthetic sub-questions/answers isn’t audited.**

Decompositions can be inconsistent or non-minimal, and recomposition relies on an LLM to produce sub-answers. The order-preserving argument assumes a particular noise structure, but we don’t see concrete diagnostics. Although the results demonstrate the effectiveness of the pipeline, we still need more diagnostics like sub-answer error rates; factor-label drift across seeds/images; sensitivity of GRPO to sub-reward noise. Right now, reliability is assumed rather than demonstrated.

**4. Model ablation is narrow.**

Results are limited to Qwen2.5-VL-7B. It’s unclear whether the gains persist across families (and weaker/stronger baselines) or scale with model size. A light grid—e.g., a smaller and a larger open model, plus a different family—would make the real-world applicability much clearer.

[1]. Self-Instruct: Aligning Language Models with Self-Generated Instructions

[2]. WizardLM: Empowering large pre-trained language models to follow complex instructions

[3]. Automatic Instruction Evolving for Large Language Models

[4]. ReST-MCTS*: LLM Self-Training via Process Reward Guided Tree Search

[5]. Self-Rewarding Language Models

**Questions:**

1. Decomposition quality: What is the inter-run agreement for factor labels/sub-questions on the same seed? Any filtering or self-consistency checks?


2. Mixture strategy: In factor-level mixing, how often do cross-dataset factors actually co-occur in recomposed questions? Can you provide some domain drift failure cases?


3. Factor generation: When MLLM decomposes the main question and generate the subquestion with the Factor, how to make sure the number of the generated Factor is within a reasonable range?

---

> ### Author Response · Authors · 2025-11-22
>
> We thank the reviewer for the constructive feedback and are glad to see their acknowledgement of our practical benefits, the effectiveness of process rewards, and our contributions on data mixing. We address each comment and question individually below.
>
> **W1. Positioning and related work.** We thank the reviewer for pointing out the relevant prior work on text-based instruction synthesis. We have revised the Related Work section accordingly (see line 90).
>
> > Prior work on automatic instruction generation [1], refinement, and evolutionary methods [2,3] increases the complexity of synthetic data for text reasoning. While these methods mainly search for reasoning trajectories in text space, we aim to analyze and augment the seed dataset by automatically detecting reasoning components grounded in visual features. Unlike approaches that rely on hand-crafted heuristics [2] or strong pretrained language models [3], COGS extracts component groups from the seed data and uses them to customize the dataset for the target task. In parallel with generalist MLLMs, specialist models have been introduced to target these domains more directly, prioritizing structured text extraction, numeric grounding, and compositional reasoning. New benchmarks and data-synthesis methods based on human-defined heuristics have followed, and specialist models have been trained accordingly.
>
> **W2. Broader comparisons.** Thank you for the suggestion. To better situate our contribution, we conducted an experiment using a representative RL-based data-generation approach as suggested by you. Specifically, we followed [5] to implement self-instruction creation for data synthesis and trained models on the resulting outputs. The performance achieved was 51.17%. This comparison helps clarify how our method differs from, and improves upon, data synthesis pipelines that rely on whole-instruction generation and self-rewarding with LLM-as-judge.
>
> We also agree that CMTS [4] is an effective approach for strengthening process-reward guidance, but it is orthogonal to our focus on data construction: CMTS can be integrated with COGS-generated data to further enhance process supervision. We have added this point to the discussion and highlighted it as promising future work (line 491).
>
> **W3. Reliability of subquestions and answers.**
>
> (1) Factor–label drift. Statistically, we analyzed three additional factor pools with different seeds that each decomposed independently from the same dataset. Compared with the pool used in the main experiment, only 0.42%, 0.36%, and 0.37% of subquestions introduced distinct factors, proving the minimal drift across seeds. Conceptually, during data generation, these factors are recomposed into complex questions by LLMs rather than following fixed templates. Therefore, as long as the factors capture primary patterns and are sampled with sufficient coverage, the synthesized data covers the task space and effectively supports training. We did not find the factor–label drift across seeds to be problematic.
>
> Factors do vary across questions and images, so a reasonable amount of seed data is needed to cover the space. In our main experiment, we show the improved performance with only 33% data as the seed set.
>
> (2) Sub-answer error rates: We randomly sampled 50 complex questions and examined their subquestions. Only 4.71% of the subquestions were incorrect. PRM-max empirically resolved these errors.
>
> (3) Concrete explanations for the ‘noise structure’. We use PRM-max to cover 2 types of noise:
>
> - Diverse reasoning paths for a single complex question. For example, a question asking for the value of the largest portion in a pie chart can be solved by two chains: Chain 1: identify all values in the pie chart → rank the values → select the highest value. Chain 2: find the largest portion by visual observation → identify its value. We generate complex questions by recomposing, meaning we have subquestions first and then the complex question, so we can cover only one reasoning chain per complex question. PRM-max avoids suppressing remaining chains if the final answer is correct.
> - Subquestions with incorrect answers are filtered out during recomposition. We prefer correct labels over complete but incorrect labels for complex questions and their subquestions. This means a small portion of subquestions attached to a complex question may be filtered out, so the subquestions may not fully cover the entire complex question. Although this effect is minor, we still sort with PRM-max.
>
> (4) Sensitivity of GRPO to sub-reward noise:
> Our ablation compares sensitivity by evaluating models trained with StandardRM, ProcessRM-sum, and ProcessRM-max. ProcessRM-sum is 0.61 lower than StandardRM, indicating that GRPO is relatively sensitive. While StandardRM improves the base model, ProcessRM-max provides a further boost, showing the effectiveness of our framework, especially when the complex question is too difficult to answer directly.

---

> > ### Author Response · Authors · 2025-11-22
> >
> > **W4. More ablation results.**
> >
> > In our initial submission, we selected Qwen2.5-VL-7B as the base model for two reasons:
> > (1) we chose the 7B scale based on our available computational resources, and
> > (2) Qwen2.5-VL-7B was the state-of-the-art open-source model for artificial-image QA at the time and is widely used in prior post-training work. For example, many previous studies evaluate exclusively on LLaMA or Qwen models [5–8].
> >
> > We appreciate the reviewer’s suggestion. We conducted additional experiments examining (i) model size within the same family (Qwen2.5-VL) and (ii) a different model family at the same 7B scale (LLaVA-1.5-7B). Specifically, we fine-tuned Qwen2.5-VL-3B and LLaVA-1.5-7B using the same synthesized data and RL settings as in the main experiments, and evaluated their resulting performance.
> >
> > We observe that COGS consistently improves the base model across architectures and scales. The results are shown below:
> > |              | Overall (Base) | Overall (+COGS) |
> > |--------------|:--------------:|:---------------:|
> > | Qwen2.5-VL-3B |     36.22%     |     38.68%      |
> > | LLaVA-v1.5-7B |     13.47%     |     22.04%      |
> > | Qwen2.5-VL-7B |     47.36%     |     52.02%      |
> >
> > **Q1. Decomposition quality.** W analyzed three additional factor pools, each generated through independent decompositions of the same dataset. Compared with the pool used in the main experiment, only 0.42%, 0.36%, and 0.37% of sub-questions introduced distinct factors, indicating very high inter-run agreement and stable decomposition quality.
> >
> > During data generation, these factors are recomposed into complex questions rather than reused as fixed templates. Therefore, as long as the factors capture the primary reasoning patterns and are sampled with sufficient coverage, the synthesized data should span the task space and effectively support training.
> >
> > **Q2. Mixture strategy.** We randomly sampled 500 recomposed questions generated from a mixed factor pool combining MMC and ChartQAPro. Among these, 286 (57.2%) include factors sampled from both datasets, 106 (21.2%) use ChartQAPro-only factors, and 108 (21.6%) use MMC-only factors. This confirms that the mixed factor pool naturally produces a diverse set of cross-dataset compositions.
> >
> > Since both seed datasets come from the chart domain, the factors overlap substantially: 97.12% of subquestions in MMC share the factor with ChartQAPro, and 88.08% of ChartQAPro with MMC. We did not observe significant domain-drift failures.
> >
> > **Q3. Factor generation.** We provide a fixed set of in-context examples in the decomposition prompt to ensure consistent factor granularity across all questions. Our in-context examples include:
> >
> > > (Question: How many times has the satisfied rate been above 25?) = (identification: satisfied rate of each year) + (comparison: each instance > 25) + (count: number of instances where satisfied rate > 25)
> > > (Question: Is the following statement True or False? gen x have experienced a steeper population increase than baby boomers did between 1990 and 2015.) = (identification: gen x’s population increase curve) + (identification: baby boomers increase curve) + (comparison: Which one has a steeper curve) + (fact_checking: Given the finding from the previous step, is the statement true?)
> >
> > Based on statistics from the 33% ChartQAPro seed split, we observe the following distribution of decomposition depth:
> > - 41.76% of questions decompose into 3 sub-questions,
> > - 22.53% decompose into 4 sub-questions,
> > - The majority fall into this 3–4-step range.
> > The maximum number of sub-questions is 7, which appears in only 0.92% of seed questions.
> >
> > Here is an example for 7 subquestions:
> > (Question: calculate the difference between the total remittances from uae in february 2017 and february 2018, then subtract the difference between the remittances from the uk in the same months. what is the result?) =
> > (identification: total remittances from UAE in Feb 2017)
> > + (identification: total remittances from UAE in Feb 2018)
> > + (calculation: difference between UAE remittances)
> > + (identification: total remittances from UK in Feb 2017)
> > + (identification: total remittances from UK in Feb 2018)
> >  + (calculation: difference between UK remittances)
> > + (calculation: result of subtracting UK remittance difference from UAE remittance difference)
> > This example illustrates controllable granularity.
> >
> > ---
> >
> > We again thank the reviewer for the constructive feedback and insightful questions. We greatly appreciate the suggestions on related work for broader comparisons. We hope our clarifications and experiments have addressed your concerns and helped illustrate the rationale behind our framework design. We are happy to provide any additional details if needed, and kindly hope you to consider revisiting your score in light of these clarifications.

---

> > > ### Author Response · Authors · 2025-11-22
> > >
> > > References
> > >
> > > [1] Self-Instruct: Aligning Language Models with Self-Generated Instructions
> > >
> > > [2] WizardLM: Empowering large pre-trained language models to follow complex instructions
> > >
> > > [3] Automatic Instruction Evolving for Large Language Models
> > >
> > > [4] ReST-MCTS*: LLM Self-Training via Process Reward Guided Tree Search
> > >
> > > [5] Self-Rewarding Language Models
> > >
> > > [6] SALMON: Self-Alignment with Instructable Reward Models
> > >
> > > [7] Chart-R1: Chain-of-Thought Supervision and Reinforcement for Advanced Chart Reasoner
> > >
> > > [8] MMC: Advancing Multimodal Chart Understanding with Large-scale Instruction Tuning

---

> ### Author Response · Authors · 2025-11-26
>
> Dear reviewer jHKn,
> Thank you for your constructive review and insightful suggestions. We have added clarifications and additional checks that directly address your comments, and have updated the manuscript accordingly. Please let us know if any further information would be helpful. Thank you again for your time and effort, and enjoy the Thanksgiving weekend.

---

### Official Review · Reviewer_ceC7 · 2025-10-30

**Soundness:** 2
**Presentation:** 3
**Contribution:** 2
**Rating:** 4
**Confidence:** 4

**Summary:**

This paper presents COGS, a framework for augmenting the reasoning capabilities of multimodal large language models (MLLMs) in domains lacking large annotated datasets, such as charts and webpages. The key idea is to decompose seed questions into primitive factors, and then systematically recompose these factors with new images to generate diverse, compositional training data.

**Strengths:**

COGS factorizes a small seed set into reusable perception/reasoning factors, recomposes them with new images to create diverse, grounded QA pairs , and uses the factor structure to supply process-level rewards, yielding richer supervision than final-answer matching alone.

**Weaknesses:**

1. The authors state that the framework *does not require ground-truth subquestion answers*, but this appears to hold only for the *seed set*: during data synthesis, each sample includes LLM-generated sub-answers that are then used for process-level supervision in RL. If these pseudo-labels are noisy, errors may accumulate and be amplified through factor-level recomposition.

2. The manuscript does not isolate the factor pool’s *direct* contribution, emphasizing cross-dataset ablations instead. Clarifying its substantive, incremental value, and distinguishing it from direct decomposing/recomposing method, would make the contribution more transparent.

3. The title is *COMPOSITION-GROUNDED **INSTRUCTION SYNTHESIS**  FOR **VISUAL REASONING***, yet the manuscript does not clearly foreground *instruction synthesis* and *visual reasoning*; it predominantly centers on chart QA. The related-work section likewise emphasizes task descriptions rather than surveying these two threads. Moreover, while the introduction mentions tables and documents, the experiments are confined to a relatively narrow setting. Chart QA can serve as a proxy for visual reasoning, but the heavy focus on charts leaves the broader visual-reasoning aspect underdeveloped.

4. The approach is mainly applicable when a task admits a reliable decomposition into subquestions; for complex cases that are not readily decomposable, its utility is limited.

5. Minor editorial issues: on lines 71 and 205, ***Grouped Rollout Policy Optimization*** should be ***Grouped Relative Policy Optimization***; on line 184, the enumerator should be ***(i)***.

**Questions:**

1. Does the manuscript describe any validation mechanism for the generated subquestion–answer pairs to prevent intermediate errors from propagating and compounding through the pipeline?

2. Can the authors provide empirical results or error analyses examining the quality/diversity of the generated synthetic data?

3. Is there any analysis of how the choice/size of the initial seed set $\mathcal{Q}^0$ affects performance or coverage? For instance, how does model generalization degrade when the seed is minimal or unrepresentative?

---

> ### Author Response · Authors · 2025-11-22
>
> We thank the reviewer for the constructive feedback. We address each comment and question individually below. We also appreciate the reviewer’s follow-up discussion, and please feel free to let us know if any point would benefit from further clarification.
>
> **W1. Pseudo-labels.** Thanks for suggesting this point for discussion. By “not requiring ground-truth answers,” we mean that our framework does not rely on human-labeled sub-questions, final questions, or answers. During data generation, our synthesis pipeline produces synthetic labels at the recomposition stage.
>
> You are correct that, in principle, label errors in the synthesis process could accumulate. In our framework, primitive questions are intentionally simple and have very high accuracy when answered by the model. Because recomposition is fully deterministic and does not modify the primitive predictions, the composed instructions faithfully preserve the underlying correct answers. This is similar in spirit to a wide range of prior work on LLM-based data synthesis [1–8].
>
> In addition, we explicitly employ multi-round verification to improve the quality of LLM-generated data. This ensures that the vast majority of synthesized questions retain correct answers. In practice, we did not observe noisy pseudo-labels to be an issue, neither in our experiments nor in the related literature.
>
> **W2. Contribution of the factor pool.** As suggested by the reviewer, we have implemented a direct decompose/recompose pipeline without using the factor pool. Specifically, we directly sample sub-questions from decompositions and recombine them into new questions for a new image. Although the LLM can adapt each composed question to the target image, the resulting dataset is noticeably less diverse and less representative. A model fine-tuned on this dataset achieves 48.78%, whereas fine-tuning on data generated with the factor pool achieves 52.02%. This demonstrates that the factor pool plays an important role in improving data coverage and downstream performance.
>
> **W3. Title, related work, and scope.** We appreciate the reviewer’s comment regarding the breadth of our title. We would like to clarify the following:
>
> (1) Scope of tasks.
> Our experiments include both ChartQA and WebGUI QA, two equally important tasks used to evaluate the effectiveness of COGS. Both domains fall under artificial-image visual reasoning and require multi-step reasoning over structured scenes. Together, they represent the scope we aim to address.
>
> (2) Explanation of the title.
> Our choice of Composition-Grounded Instruction Synthesis for Visual Reasoning reflects that our method conceptually uses structure-aware representations to achieve task-agnostic competence. The proposed instruction-synthesis mechanism is designed to support **multi-step, compositionally grounded visual reasoning over structured images**, rather than targeting charts specifically. The framework operates on perception-level and reasoning-level cues extracted from artificial images, which motivated presenting the work under a general visual-reasoning umbrella.
>
> While chart-based reasoning constitutes a substantial portion of our experiments, largely because it offers a well-established and controlled testbed --- we do not restrict ourselves to charts. As detailed in Section 4.2 (Lines 375–407), we also evaluate on WebGUI QA, a visually and semantically distinct task involving complex UI layouts. This setting requires interpreting functional groupings, spatial hierarchies, and symbolic components, and it demands the same multi-step, compositional reasoning that our method is designed to enable. Together, ChartQA and WebGUI QA serve as two distinct instantiations of structured artificial-image visual reasoning, each comprising a substantial portion of our experiments and jointly demonstrating the cross-domain generality of our approach.
>
> If the current framing appears broader than the empirical range, we are open to refining the title for added clarity. Possible alternatives include:
>
> - Composition-Grounded Instruction Synthesis for Structured Visual Reasoning
> - Instruction Synthesis for Compositional Reasoning over Structured Visual Scenes
> - Compositional Instruction Synthesis for Visual Reasoning in Information-Rich Images
>
> All three options remain faithful to our core methods and more explicitly reflect the domains evaluated in our experiments. We would be glad to hear the reviewer's suggestions on which formulation best describes the paper’s scope.

---

> > ### Author Response · Authors · 2025-11-22
> >
> > (3) Related work.
> > Regarding the organization of related work, we cover benchmarks, data/instruction-synthesis approaches, and specialist models for both chart and web UI question answering. We structured this section by task type to maintain readability and parallelism. Specifically:
> >
> > - Lines 93–99: benchmarks evaluating perception and reasoning in charts
> > - Lines 99–101: chart data-synthesis approaches
> > - Lines 122–124: web benchmarks and methods (mirroring the structure used for charts)
> >
> > **W4. Decompositionality.** We offer a solution for complex question-answering tasks, based on the observation that complex questions can typically be decomposed even when their structure is not explicitly stated. We would like to emphasize that compositionality is broadly applicable across many important problem settings and has been leveraged in prior work on text generation [9–10], visual reasoning [11–12], and action planning [13–14], using either explicit or implicit forms of concept decomposition.
> >
> > **W5. Typos.** We thank the reviewer for pointing out these typos. We have corrected them in the updated manuscript.
> >
> > **Q1. Answer validations.** The use of LLMs as answer verifiers in data synthesis is now widely adopted, so we did not present it as a contribution of this work. In line with prior methods, we use an LLM to verify answers for both sub-questions and composed complex questions. We perform multi-round verification and retain only those questions whose answers reach agreement across rounds. This ensures that the majority of synthesized questions have correct corresponding answers. For additional details, please refer to our response to Weakness W1.
> >
> > **Q2. Error analysis of synthetic data.** We randomly sampled 50 complex questions for manual inspection. The final-answer accuracy was 86%, and the sub-question accuracy was 95.29%. Both the composed questions and their sub-questions are high-quality and effectively support RL training. Using PRM-max, which takes the maximum of the final-answer and sub-question rewards, further improves the robustness of this supervision.
> >
> > **Q3. Seed set studies.** We conducted data synthesis and RL training using seed subsets of 1%, 15%, 25%, and 33% of the original ChartQAPro dataset. For fair comparison, we fixed the test set at 67% and sampled seed questions only from the remaining 33%. Overall, the performance improves steadily with larger seed sets. We will add this ablation to the paper:
> >
> > - 1%: 48.95%
> > - 15%: 50.45%
> > - 25%: 51.57%
> > - 33%: 52.02%
> >
> > We observed that very small seed sets are less representative, but using a reasonable subset, such as 33%, already yields substantial performance gains.
> >
> > ---
> >
> > We again thank the reviewer for the constructive feedback and insightful questions. Please feel free to let us know your preference regarding the title. We hope that our clarifications and additional experiments have addressed your concerns and helped clarify the rationale behind our framework design. We are happy to provide any further details if needed, and we kindly ask the reviewer to consider revisiting their score in light of these explanations.
> >
> > References
> >
> > [1] Self-Rewarding Language Models
> >
> > [2] SALMON: Self-Alignment with Instructable Reward Models
> >
> > [3] RLAIF vs. RLHF: Scaling Reinforcement Learning from Human Feedback with AI Feedback
> >
> > [4] Process-based Self-Rewarding Language Models
> >
> > [5] J1: Incentivizing Thinking in LLM-as-a-Judge via Reinforcement Learning
> >
> > [6] Chart-R1: Chain-of-Thought Supervision and Reinforcement for Advanced Chart Reasoner
> >
> > [7] ChartMoE: Mixture of Diversely Aligned Expert Connector for Chart Understanding
> >
> > [8] Harnessing Webpage UIs for Text-Rich Visual Understanding
> >
> > [9] Decompose, analyze and rethink: solving intricate problems with human-like reasoning cycle
> >
> > [10] Least-to-Most Prompting: Enabling Complex Reasoning in Large Language Models
> >
> > [11] What Makes a Maze Look Like a Maze?
> >
> > [12] Visual Programming: Compositional Visual Reasoning Without Training
> >
> > [13] Language Models as Zero-Shot Planners: Extracting Actionable Knowledge for Embodied Agents
> >
> > [14] Do As I Can, Not As I Say: Grounding Language in Robotic Affordances

---

> > > ### Author Response · Authors · 2025-11-26
> > >
> > > Dear reviewer ceC7,
> > > Thank you for the constructive reviews. We added clarifications and additional checks that directly address your comments. We look forward to your feedback on our clarifications and experiments, as well as your preference regarding the title. We are happy to incorporate any changes you suggest. Thank you again for your time, and enjoy the Thanksgiving weekend.

---

### Official Review · Reviewer_hVuu · 2025-11-05

**Soundness:** 3
**Presentation:** 3
**Contribution:** 4
**Rating:** 6
**Confidence:** 3

**Summary:**

This paper introduces COGS, a data-efficient framework for enhancing visual reasoning capabilities in multimodal large language models  for artificial image domains like charts and webpages. The key innovation is decomposing seed questions into primitive perception and reasoning "factors," then systematically recombining these factors with new unlabeled images to generate synthetic training data. Each generated question includes subquestions and intermediate answers, enabling reinforcement learning with process-level rewards. The authors evaluate COGS primarily on chart reasoning (ChartQAPro, MMC-Bench) and webpage understanding (VisualWebBench), demonstrating substantial improvements over baseline models, with particularly strong gains on reasoning-heavy and compositional questions.

**Strengths:**

- Novel compositional approach: The factor decomposition and recomposition strategy is intuitive and well-motivated, providing a principled way to scale up training data from limited seed examples while maintaining diversity and complexity.

- Strong empirical results: COGS achieves meaningful improvements across multiple benchmarks (52.02% on ChartQAPro vs. 47.36% base model; 88.04% on VisualWebBench vs. 85.65% base), outperforming both general-purpose models and domain-specific baselines.

- Transferability across datasets: The factor-level mixture experiments (Section 4.1.2) demonstrate that COGS induces generalizable reasoning capabilities rather than dataset-specific overfitting, with factor-level mixing outperforming data-level mixing.

- Theoretical contribution: Proposition 3.1 provides valuable theoretical insight into why ProcessRM-max preserves policy ordering while ProcessRM-sum does not, backed by empirical validation.

- Comprehensive evaluation: The paper includes thorough ablations (reward models, question complexity, factor types) and extends beyond charts to webpages, demonstrating generalizability.

**Weaknesses:**

- The paper focuses exclusively on artificial image domains (charts, webpages). It's unclear whether this approach would transfer to natural images or other multimodal reasoning tasks. The restriction to domains with "abundant unlabeled images" may limit applicability.

- Dependency on high-quality decomposition: The entire framework relies on an MLLM's ability to accurately decompose questions into factors. The paper doesn't thoroughly analyze decomposition quality or failure modes. What happens when decomposition is incorrect or incomplete?

- Limited baseline comparisons: While the paper compares against several data synthesis approaches, it doesn't compare against other compositional reasoning methods or more sophisticated prompting techniques (e.g., self-consistency, tree-of-thoughts).

**Questions:**

Can factors learned from one domain (e.g., charts) transfer to another domain (e.g., webpages) without redecomposition? Would a shared factor pool improve multi-domain performance?

---

> ### Author Response · Authors · 2025-11-22
>
> We thank the reviewer for the constructive feedback and are glad that they appreciate our novel approach, data-mixing strategy, and theoretical contributions. We also appreciate the reviewer’s recognition of our comprehensive evaluation and strong empirical results. We address the comments and questions below.
>
> **W1. Domain applicability.** Our method is intentionally scoped to the artificial-image domain. This domain is important in real applications such as data-visualization interfaces (e.g., analytical dashboards), enterprise reporting systems, and GUI automation, and has been widely studied in prior work [1–5]. Existing work for this domain typically relies on specialist models and benchmarks designed for a single task family (e.g., chart QA only, document QA only, or UI QA only). In contrast, our approach provides a more general instruction-synthesis framework that unifies compositional reasoning across multiple structured artificial-image domains, already extending beyond existing task-specific pipelines.
>
> **W2. Decomposition quality.** We thank the reviewer for raising this point. In principle, three types of failure modes may occur:
>
> 1. Incorrect coverage.
> The sub-questions derived from a complex question may not perfectly reconstruct the original question. This does not harm our pipeline because recomposition samples factors to create entirely new questions grounded in new images, independent of the seed question. Training uses only these generated questions and labels, so this failure mode does not introduce incorrect supervision. We have also manually inspected decompositions and did not observe such cases.
> 2. Correct but incomplete.
> For example, “What is the difference between A and B?” might decompose into “value(A)” and “difference(B, A)” rather than the canonical composition of “value(A), value(B), difference(A, B).” This is acceptable, as these outputs still provide structure for recomposition and ensure that the synthesized complex questions span the intended task space.
> 3. Instruction non-compliance.
> The model may occasionally deviate from the provided decomposition instructions. We mitigate this through curated in-context examples, and manual inspection found no such violations.
>
> In practice, we found that these failures are rare and have no observable impact on training performance. For this reason, we did not highlight them in the initial submission. We are happy to include the above discussion in the revision for completeness.
>
> **W3. Baselines.** In the paper, we compared against inference-time decomposition as the most directly relevant compositional-reasoning baseline (see Additional Variants, lines 323–326 of the initial submission, and Decompositional CoT in Table 1). We thank the reviewer for suggesting additional representative baselines. We therefore evaluate Tree of Thoughts and Self-Consistency, using Qwen2.5VL-7B as the base model on ChartQAPro.
>
> For Self-Consistency, we sample 10 responses per query and use the majority vote as the final answer; the overall accuracy is 47.22%. For Tree of Thoughts, we follow the official implementation (https://github.com/princeton-nlp/tree-of-thought-llm) and extend it to support image input; the resulting accuracy is 44.44%.
>
> We conjecture that although Tree of Thoughts is more effective for search-heavy tasks such as solving puzzles or finding proofs, it is not well suited for settings where reasoning depends on visual recognition combined with largely deterministic functional operations. Moreover, for both inference-time improvement techniques, performance remains fundamentally constrained by the capability of the underlying base model, while our framework synthesizes new data for improving the model.

---

> > ### Author Response · Authors · 2025-11-22
> >
> > **Q1. Transfer of factors across different domains.**
> > We did not observe positive transfer between the two domains evaluated in the paper (Charts and Web), although we believe cross-domain transfer may emerge once more domains are incorporated. The primary limitation is that both the visual features and the question distributions vary substantially between the two domains. Because factors reflect the underlying task distribution, factors learned in one domain may not align well with those in another.
> >
> > Manual inspection supports this: primary reasoning factors in the Web domain are “content inspection,” “spatial relations,” and “status,” whereas chart-domain factors are primarily “calculation” and “fact checking.” Although some factors, such as “counting” and “comparison,” appear in both domains, they constitute only a small fraction of WebQA (6% and 1.9%, respectively).
> >
> > In our experiments, chart-domain factors did not provide measurable gains on WebGUI QA. When we used chart-derived factors to compose questions for web images, the resulting model achieved 84.21% on the WebQA test set, whereas a model trained on WebGUI-derived factors reached 88.04% (The base model achieves 85.65%). This confirms that domain mismatch limits factor transferability.
> > That said, our paper demonstrates that factors are broadly reusable within a domain (e.g., across multiple chart datasets), where they yield consistent and significant improvements (see Table 2).
> >
> > ⸻
> >
> > We again thank the reviewer for the constructive feedback and insightful questions. The suggested prompting baselines were very helpful, and we have incorporated them into the main results table. We hope that our clarifications and additional experiments address the reviewer’s concerns and help illustrate the rationale behind our framework design. We are happy to provide any further details if needed, and we kindly ask the reviewer to consider revisiting their score in light of these explanations.
> >
> > References
> >
> > [1] Chart-R1: Chain-of-Thought Supervision and Reinforcement for Advanced Chart Reasoner
> >
> > [2] ChartMoE: Mixture of Diversely Aligned Expert Connector for Chart Understanding
> >
> > [3] Harnessing Webpage UIs for Text-Rich Visual Understanding
> >
> > [4] Pix2Struct: Screenshot Parsing as Pretraining for Visual Language Understanding
> >
> > [5] ScreenSpot-Pro: GUI Grounding for Professional High-Resolution Computer Use

---

> > > ### Author Response · Authors · 2025-11-26
> > >
> > > Dear reviewer hVuu,
> > > Thank you for the constructive reviews and helpful discussions. We added clarifications and additional checks that directly address your comments. Please do not hesitate to let us know if there is anything we can provide. Thanks again for your review effort, and enjoy the Thanksgiving weekend.

---

### Author Response · Authors · 2025-11-22
**Manuscript Updates**

Following the reviewers’ feedback, we have addressed the comments through follow-up clarifications and incorporated selected suggestions into the manuscript. The updates include:

- Added two additional prompting strategies—Self-Consistency and Tree of Thoughts—to the main results table as baselines (thanks to reviewer hVuu).
- Revised the Future Work section to discuss integrating search algorithms with COGS-synthesized data for stronger reward guidance (thanks to reviewer jHKn).
- Added the ablation study on seed questions size to section 4.3 (thanks to reviewer ceC7).
- Expanded the Related Work section to better situate our contribution relative to prior work on general data synthesis (thanks to reviewer jHKn).
- Added the ablation study on base model size and family in appendix C (thanks to reviewer jHKn).
- Corrected all identified typos (thanks to reviewer ceC7).

We thank the reviewers again for their constructive feedback and helpful suggestions.

---

### Author Response · Authors · 2025-12-01
**Rebuttal Summary for the Area Chair**

We want to thank the reviewers again for their recognizing our **strength** on:

**1. Novelty (hVuu, ceC7, jHKn, dvz5)**.

&emsp; &emsp; (1) Reviewers recognize our key idea that decomposing each seed question into primitive perception and reasoning factors, and systematically recomposing the factors with new images to generate large collections of synthetic question-answer pairs is novel.

&emsp; &emsp; (2) Reviewers agree that the use of structured synthetic data for process-level rewards provide richer supervision and error analysis than final-answer matching alone. It’s effective and well-motivated.

**2. Process reward design (hVuu, jHKn)**. Both theoretical rationale and experimental evidence support the design.

**3. Effective data mixing strategy (hVuu, jHKn)**. Factor-level composition yields a practical dataset mixture that captures reusable structure.

**4. Generalizability and comprehensive analysis (jHKn)**. Broad evaluation across charts and web GUIs in the artificial image domain, plus thorough ablations on reward models, subquestions, and factor types.

**5. Empirical strength (hVuu)**. Strong results compared with baselines.

---

> ### Author Response · Authors · 2025-12-01
> **We addressed each reviewer’s comments and questions one by one in our responses. Here is a brief overview:**
>
> ### [**Reviewer hVuu**](https://openreview.net/forum?id=FnF3UjiN11&noteId=aP6o7BodSY)
>
> **W1. Domain applicability**: The reviewer commented on the applicability of our framework to other vision domains, such as natural images. We clarified that the perceived limitation stems from a misunderstanding of our intended scope: our method is purposefully designed for structured artificial images (e.g., charts, web UIs), a practically important and widely studied domain.  Our framework is in fact broader than most prior work that focused only on charts or documents.
>
> **W2. Decomposition quality**: COGS produces high-quality decompositions, as validated by our manual checks. We also analyzed potential failure modes and show that their impact is limited within our pipeline.
>
> **W3. Additional baselines**: We added Self-Consistency and Tree of Thoughts to the main results table. Our method **remains the strongest**.
>
> **Q1. Factor transfer across domains**: We have included cross-domain (charts to web) transfer results and explained factor-distribution differences across domains. Because distributions differ substantially across domains, strong factor transfer between them is not a well-posed requirement. Our paper focuses on cross-dataset factor transfer within the same domain.
>
> &nbsp;
> ### [**Reviewer ceC7**](https://openreview.net/forum?id=FnF3UjiN11&noteId=u2krcBz7Ej)
>
> **W1. Answer validation for pseudo-labels**: We detailed the verification mechanism for both subquestions and complex questions in the original submission. Please refer to [this response](https://openreview.net/forum?id=FnF3UjiN11&noteId=u2krcBz7Ej) for details.
>
> **W2. Contribution of the factor pool**: We compared against a direct decompose-and-recompose variant as suggested by the reviewer. The results showed that the factor pool ensures diversity to directly boost performance.
>
> **W3. Title, related work, and scope**: The impression that the work centers on chart QA likely stems from the Web UI experiments being overlooked. Our submission reports comprehensive experiments on both charts and web GUIs, consistent with our stated scope, and our related work already covers this breadth. We also proposed more precise title options.
>
> **W4. Decompositionality**: The reviewer’s observation may stem from limited exposure to established work on compositionality. We clarify that task and concept decomposition is prevalent across domains, as evidenced by prior work in text generation, visual reasoning, and action planning.
>
> **Q1/Q2. Pipeline validation and error analysis**: We describe the filtering of incorrect subanswers in our pipeline. Our manual analysis shows a 4.71% subanswer error rate, which is mitigated by PRM-max.
>
> **Q3. Seed-set**: We include an ablation on the size of the seed question set. Please refer to [this response](https://openreview.net/forum?id=FnF3UjiN11&noteId=i0vst1pm8W). This has also been incorporated in lines 486 to 497 of the updated manuscript.

---

> ### Author Response · Authors · 2025-12-01
>
> ### [**Reviewer jHKn**](https://openreview.net/forum?id=FnF3UjiN11&noteId=MSZFgmo5nc)
>
> **W1. Positioning and related work**: We expanded related work to include broader data-synthesis methods according to the reviewer. Please refer to lines 90-100 of our updated manuscript.
>
> **W2. Broader baselines**: We tested the reviewer-suggested data-synthesis baseline. Our method remains the **strongest**.
>
> **W3. Reliability of subquestions and answers**: We addressed the reviewer’s concern by reporting a high inter-run agreement for decomposition and systematically explaining why PRM-max improves robustness of subquestions/answers. The initial submission already includes empirical comparisons of PRM-max to StandardRM and PRM-sum.
>
> **W4. Base model size and family**: We added a size-and-family grid (Qwen2.5-VL-3B, LLaVA-1.5-7B) and observed consistent gains with COGS, which addressed the reviewer's question on whether gains persist across families or model sizes. We also incorporated this experiment in Appendix C of our manuscript.
>
> **Q1. Decomposition robustness**: The reviewer asked about the inter-run agreement of question decomposition. Our quantitative analysis answers this question by showing a high inter-run agreement, indicating the high quality and robustness of COGS decomposition.
>
> **Q2. Factor-level mixture is a real mix**: The reviewer asked about the co-occurrence of cross-dataset factors in the synthetic data generated from factor-level mixing. In cross-dataset mixing, 57.2% of recomposed questions include factors from both datasets, indicating high co-occurrence.
>
> **Q3. Factor generation details**: We clarify that granularity is stabilized with fixed in-context examples and that decomposition quality is sufficient. Therefore, the generated factor is within a reasonable range.
>
> &nbsp;
> ### [**Reviewer dvz5**](https://openreview.net/forum?id=FnF3UjiN11&noteId=mih2NuAjLs)
> **W1. Evaluation protocol and image from the seed data**: The reviewer misinterpreted our image usage for data synthesis and evaluation. We clarify that sources are **separated** and the image pools are **disjoint**. All synthesized questions are grounded in ChartQA-train images instead of ChartQAPro.
>
> **W2. Compute overhead of LLM-as-a-judge**:  The reviewer’s hypothesis of significant training overhead from LLM-as-a-judge for process reward is not supported by our experiments. Reward inference accounts for only 26% of total training time and is not the bottleneck. Please find detailed computation analysis in [this response](https://openreview.net/forum?id=FnF3UjiN11&noteId=mih2NuAjLs).

---

> ### Author Response · Authors · 2025-12-01
>
> We unfortunately did not receive the reviewers’ feedback before the ICLR rebuttal policy switch. Although the reviewers have not replied and may not be able to do so, our responses have addressed all of their comments. Given this, we hope the area chair can evaluate our experiments and clarifications on the same footing. Sincerely appreciate your time and effort!

---

### Meta-Review · Area_Chair_SKsz · 2026-01-14

**Summary:**

The paper proposes COGS, a framework for synthesizing instructional data for visual reasoning by decomposing seed questions into perception and reasoning factors and recomposing them with new images. The reviewers recognized the novelty of the proposed approach and the theoretical motivation behind the process reward design. However, the initial reviews (Scores: 6, 4, 4, 4) highlighted significant concerns regarding the breadth of baselines, the robustness of the synthesis pipeline, potential data leakage, and the limited variety of base models tested. My decision to recommend acceptance is informed by the authors' exceptionally comprehensive rebuttal, which provided new experimental evidence (new baselines, model ablations, and error analysis) that directly resolved the major technical concerns raised by the reviewers.

**Reviewer Concerns:**

The authors provided a detailed rebuttal. Based on their official responses and manuscript revisions, most concerns on more baselines, model generalization, decomposition quality, details of method have been adequately addressed.

The remaining issue relates to domain scope; however, given the paper’s explicit scoping, this is an acceptable limitation rather than a substantive weakness.

**Reviewer Scores:**

Reviewer hVuu (Current: 6): Predicted: 6. This reviewer was already positive. The addition of the requested baselines and the clarification on failure modes strengthens the paper further.

Reviewer ceC7 (Current: 4): Predicted: 4. The authors addressed this reviewer's request for an ablation on the factor pool and provided the requested error analysis.

Reviewer jHKn (Current: 4): Predicted: 6. The authors added the related work and broader baselines for the rebuttal.

Reviewer dvz5 (Current: 4): Predicted: 6. This score appeared heavily influenced by the "data leakage" suspicion. With the authors clarifying that the image and question sets are disjoint, this major red flag is removed.

---

### Decision · Program_Chairs · 2026-01-26

Accept (Poster)